# The Kernel Perspective on Dynamic Mode Decomposition

**Efrain H Gonzalez**[*]                                                      *ehgonza@sandia.gov*
*Department of Mathematics and Statistics*
*University of South Florida*

**Moad Abudia**                                                              *abudia@okstate.edu*
*School of Mechanical and Aerospace Engineering*
*Oklahoma State University*

**Michael Jury**                                                            *mjury@ufl.edu*
*Department of Mathematics*
*University of Florida*

**Rushikesh Kamalapurkar**                                              *rkamalapurkar@ufl.edu*
*Department of Mechanical and Aerospace Engineering*
*University of Florida*

**Joel A. Rosenfeld**[†]                                                  *rosenfeldj@usf.edu*
*Department of Mathematics and Statistics*
*University of South Florida*

**Reviewed on OpenReview:** *https://openreview.net/forum?id=sIR8xV7hGl*

## Abstract

This manuscript takes a critical look at the interactions between Koopman theory and reproducing kernel Hilbert spaces with an eye towards giving a tighter theoretical foundation for Koopman based dynamic mode decomposition (DMD), a data driven method for modeling a nonlinear dynamical system from snapshots. In particular, this paper explores the various necessary conditions imposed on the dynamics when a Koopman operator is bounded or compact over a reproducing kernel Hilbert space.

Ultimately, it is determined that for many RKHSs, the imposition of compactness or boundedness on a Koopman operator forces the dynamics to be affine.

However, a numerical method is still recovered in more general cases through the consideration of the Koopman operator as a closed and densely defined operator, which requires a closer examination of the connection between the Koopman operator and a RKHS. By abandoning the feature representation of RKHSs, the tools of function theory are brought to bear, and a simpler algorithm is obtained for DMD than what was introduced in Williams et al (2016). This algorithm is also generalized to utilize vector valued RKHSs.

## 1 Introduction

Dynamic mode decomposition (DMD) has been gaining traction as a model-free method of making short-run predictions for nonlinear dynamical systems using data obtained as snapshots of trajectories. DMD aims to obtain a finite rank representation of the Koopman operator by studying its action on the full state observable (i.e. the identity function) (Rowley et al., 2009). Koopman operators over reproducing kernel Hilbert spaces (RKHSs) were studied to take advantage of infinite dimensional feature spaces to extract more information from the snapshots of a system in (Williams et al., 2015a). This perspective also enacts a dimensionality reduction by formulating the DMD method in a reproducing kernel Hilbert space (RKHS) framework and implicitly using the kernel trick to compute inner products in the high-dimensional space of observables. In

---

[*]Now at the Department of Cognitive & Emerging Computing, Sandia National Laboratories

[†]The following YouTube playlist discusses the content of this manuscript: `https://youtube.com/playlist?list=PLldiDnQu2phuB64ccOYWxeSBZxqOGg47g`

(Williams et al., 2015b), it is shown that kernel-based DMD produces a collection of Koopman modes that agrees with other DMD results in the literature.

The introduction of kernel based techniques for Koopman analysis and DMD has catalyzed a new direction for Koopmanism. The new perspective added by kernel methods is that of approximations and it is at the center of the work presented in this manuscript. Universal RKHSs, such as those corresponding to Gaussian RBFs and exponential dot product kernel functions, have the ability to approximate any continuous function over a compact subset of $\mathbb{R}^n$ to any desired accuracy up to computational precision. Moreover, when the kernel function is continuous or bounded, convergence in RKHS norm yields point wise everywhere convergence and uniform convergence over compact subsets of $\mathbb{R}^n$ (Steinwart & Christmann, 2008). This everywhere convergence stands in contrast to the perspective through the lens of ergodic methods, where convergence results only hold almost everywhere. For additional insight into the point wise everywhere convergence property described above see example 1 in Appendix A.1.

The study of dynamical systems through the Koopman formalism over RKHSs manifests as a search for functions that are close to being eigenfunctions of the Koopman operator, rather than the actual eigenfunctions. Since only a finite amount of data can be available for the study of an infinite dimensional operator, actual eigenfunctions typically cannot be computed. In fact, there is no requirement that $\mathcal{K}_F$ even has any eigenfunctions. The universality property arises in the search for Koopman "eigenfunctions," which, given a particular Koopman operator and kernel space, might not exist, since the existence of an eigenfunction depends on the Hilbert space. However, the formal equation, $\varphi(F(x)) - \lambda\varphi(x) = 0$ may still hold for some continuous function $\varphi$. If one is working over a RKHS corresponding to a universal kernel, then for any given compact set and $\varepsilon > 0$, there is a function $\tilde{\varphi} \in H$ such that $|\varphi(x) - \tilde{\varphi}(x)| < \varepsilon$ for all $x$ in that compact set, which provides for an "approximate" eigenfunction. Here we define approximate eigenfunctions as follows:

**Definition 1.** For $\epsilon > 0$ an approximate eigenfunction $\hat{\varphi}$ is a function in the Hilbert space $H$ together with $\hat{\lambda} \in \mathbb{C}$ such that $|\mathcal{K}_F\hat{\varphi}(x) - \hat{\lambda}\hat{\varphi}(x)| < \epsilon$ for all $x \in X$.

This is particularly important for DMD methods, which attempt to construct a finite rank approximation of a Koopman operator from a finite collection of observed snapshots. Note that obtaining approximate eigenfunctions as in Example 1 is not dissimilar to the objective of ergodic methods, where approximation of system invariants and eigenfunctions using time averages is sought. The existence of eigenfunctions depends on the selection of the Hilbert space, as will be shown in Section 4, and eigenfunctions may not be present even in the $L^2$ ergodic setting (Budišić et al., 2012).

The objective of this manuscript is to present the kernel perspective of Koopmanism as a distinct study from the ergodic perspective. With that goal in mind, the paper is structured in the following way:

- Section 2: Introduces reproducing kernel Hilbert spaces (RKHSs) which will be necessary for the remainder of the paper.

- Section 3: Introduces Koopman operators, their relationship to DMD, and the differences between the ergodic and kernel perspectives. Here the pointwise convergence properties in the RKHS are presented.

- Section 4: Discusses properties of Koopman operators over RKHSs. It is demonstrated that assumptions of boundedness and compactness hold for a very small collection of Koopman operators.

- Section 5: A numerical algorithm for a Koopman based DMD method is presented, which relaxes the assumptions of bounded and compact Koopman operators to densely defined Koopman operators. This yields a new theoretical foundation for the study of Koopman DMD over RKHSs. The new algorithm still relies on some of the same matrices presented in (Williams et al., 2015b).

- Section 6: Extends the developed DMD algorithm to vector valued RKHSs.

- Section 7: Provides a numerical example that compares the developed DMD method to that presented in (Williams et al., 2015b).

- Appendix: Includes examples and proofs referenced in the text as well as a few numerical experiments which show that the developed DMD algorithm has nearly identical results to that developed by (Williams et al., 2015b).

## 2   Reproducing Kernel Hilbert Spaces

**Definition 2.** A Reproducing Kernel Hilbert Space (RKHS) $H$ over a set $X$ is a Hilbert space composed of functions from $X$ to $\mathbb{C}$ such that for all $x \in X$ the evaluation functional $E_x f := f(x)$ is bounded.

Therefore, for all $x \in X$ there exists a function $K_x \in H$ such that $f(x) = \langle f, K_x \rangle$ for all $f \in H$. The function $K_x$ is the reproducing kernel centered at $x$ and the function $K : X \times X \to \mathbb{C}$ defined by $K(x, y) = \langle K_y, K_x \rangle$ is the unique kernel function corresponding to $H$ (Aronszajn, 1950). Throughout most of this manuscript, the methods used will be restricted to RKHSs of real valued functions. However, for some specific examples in Section 4, it will be more convenient to employ RKHSs of complex valued functions of a real variable.

Reproducing kernels can be equivalently expressed as realizations of inner products of feature space mappings in $\ell^2(\mathbb{N})$ (Steinwart & Christmann, 2008).

**Proposition 1.** *Given an orthonormal basis for a RKHS, $\{e_m(\cdot)\}_{m=1}^\infty \subset H$, the kernel function may be expressed as $K(x, y) = \sum_{m=1}^\infty e_m(x)\overline{e_m(y)}$, where $\Psi(x) := (e_1(x), e_2(x), \ldots) \in \ell^2(\mathbb{N})$ is called a feature map. Equivalently, given a feature mapping $\Psi : X \to \ell^2(\mathbb{N})$, there is a RKHS whose kernel function is given as $K(x, y) = \langle \Psi(x), \Psi(y) \rangle_{\ell^2}$.*

We will make repeated use of projections onto finite dimensional vector spaces arising from spans of collections of kernels centered at snapshots from a dynamical system.

**Proposition 2.** *For a collection of centers $\{x_1, \ldots, x_m\}$, the projection of a function $g \in H$ onto $\alpha = \mathrm{span}\{K_{x_1}, \ldots, K_{x_m}\}$ is given as $\arg\min_{h \in \alpha} \|h - g\|_H$. The projection can be expressed in terms of $\alpha$ as $P_\alpha g := \sum_{i=1}^m w_i K_{x_i}$ where the weights $w_i$ satisfy*

$$\begin{pmatrix} K(x_1, x_1) & \cdots & K(x_1, x_m) \\ \vdots & \ddots & \vdots \\ K(x_m, x_1) & \cdots & K(x_m, x_m) \end{pmatrix} \begin{pmatrix} w_1 \\ \vdots \\ w_m \end{pmatrix} = \begin{pmatrix} g(x_1) \\ \vdots \\ g(x_m) \end{pmatrix}. \tag{1}$$

*Proof.* The projection of a function in a Hilbert space onto a closed subspace is determined by finding the closest member of that subspace to the function being projected. The matrix equation may be established by expressing the projection as $P_\alpha g = \sum_{i=1}^m w_i K_{x_i}$, expanding $\|\sum_{i=1}^m w_i K_{x_i} - g\|_H^2$ via inner products, and setting the derivative with respect to $w := (w_1, \ldots, w_m)^T \in \mathbb{R}^m$ to zero, resulting in the system of linear equations in equation 1. $\square$

If the functions $\{K_{x_1}, \ldots, K_{x_m}\}$ are linearly independent then the Gram matrix in equation 1 is non-singular. In that case, the weights $\{w_i\}_{i=1}^m$ are unique.

**Definition 3.** A RKHS of real valued functions, $H$, over $\Omega \subset \mathbb{R}^n$, is said to be universal if for any compact $V \subset \Omega$, $\epsilon > 0$, and $h \in C(V)$, there is a function $\tilde{h} \in H$ such that $\|h - \tilde{h}\|_\infty < \epsilon$, where $C(V)$ denotes the set of continuous functions defined on $V$.

Many commonly used kernel functions satisfy this universality property, including the Gaussian RBF kernel functions and the exponential dot product kernel functions (Steinwart & Christmann, 2008).

## 3   Koopman Operators over RKHSs

The theory of Koopman operators has been long intertwined with ergodic theory, where ergodic theoretic methods justify almost everywhere convergence claims of time averaging methods to invariants of the Koopman operator. The Birkhoff and Von Neumann ergodic theorems are posed over $L^1(\mathbb{R})$ and $L^p(\mathbb{R})$ for $p > 1$, respectively (Walters, 2000). However, the invariants for Koopman operators are not always analytic or even

smooth. Hence, ergodic theorems do not give guarantees of convergence within most RKHSs, which are frequently composed of real analytic functions. Furthermore, even though ergodic theorems guarantee the existence of invariants over $L^2(\mathbb{R})$, the invariant itself is hidden behind a limiting operation via time averages (Walters, 2000). Hence, there is an expected error in the constructed invariant that stems from finiteness of data.

The objective of DMD methods is to find functions within the function space that nearly achieve eigenfunction behavior. Specifically, for a Koopman operator, $\mathcal{K}_F$, and $\epsilon > 0$, the objective is to find $\hat{\varphi} \in H$ and $\lambda \in \mathbb{C}$ for which $|\mathcal{K}_F\hat{\varphi}(x) - \lambda\hat{\varphi}(x)| < \epsilon$ for all $x$ in a given workspace. When such a function is discovered, the eigenfunction witnesses the snapshots, $x_{i+1} = F(x_i)$, as an exponential function as $\hat{\varphi}(x_{i+1}) = \lambda^i\hat{\varphi}(x_1) + \frac{1-\lambda^{i+1}}{1-\lambda} \cdot \epsilon$. If $\hat{\varphi}$ is a proper eigenfunction for $\mathcal{K}_F$, then $\epsilon$ may be taken to be zero, and $\hat{\varphi}(x_{i+1}) = \lambda^i\hat{\varphi}(x_1)$.

Ergodic methods generally yield $|\mathcal{K}_F\hat{\varphi}(x) - \lambda\hat{\varphi}(x)| < \epsilon$ only for almost all $x$ within the domain of interest. For a RKHS, the condition $|\mathcal{K}_F\hat{\varphi}(x) - \lambda\hat{\varphi}(x)| < \epsilon$ may be relaxed to $\|\mathcal{K}_F\hat{\varphi} - \lambda\hat{\varphi}\|_H < \epsilon$, since $|\mathcal{K}_F\hat{\varphi}(x) - \lambda\hat{\varphi}(x)| < C\|\mathcal{K}_F\hat{\varphi} - \lambda\hat{\varphi}\|_H < C\epsilon$, where $C > 0$ depends on the kernel function and the point $x$. If the kernel function is continuous and the domain is compact, a finite $C$ may be selected uniformly for that domain. In the special case of the Gaussian RBF kernel function and the domain being $\mathbb{R}^n$, $C$ may be taken to be 1.

**Proposition 3.** *If $\mathcal{K}_F$ is compact and the finite rank approximation of $\mathcal{K}_F$, which we will denote as $\hat{\mathcal{K}}_F$, is within $\epsilon$ of $\mathcal{K}_F$ with respect to the operator norm, and if $\hat{\varphi}$ is a normalized eigenfunction for $\hat{\mathcal{K}}_F$ with eigenvalue $\lambda$, then $\|\mathcal{K}_F\hat{\varphi} - \lambda\hat{\varphi}\|_H \leq \|\mathcal{K}_F\hat{\varphi} - \hat{\mathcal{K}}_F\hat{\varphi}\|_H \leq \|\mathcal{K}_F - \hat{\mathcal{K}}_F\| \leq \epsilon$.*

Hence, if we obtain a finite rank approximation of $\mathcal{K}_F$ that is within $\epsilon$ with respect to the operator norm, then an eigenfunction of the finite rank approximation will be an approximate eigenfunction of $\mathcal{K}_F$. This approximation is important in DMD, where the eigenfunctions are utilized to generate an approximation of the full state observable, $g_{\text{id}}(x) := x$, one dimension at a time, as outlined in Section 5.

If an accurate finite rank approximation of the Koopman operator can be obtained in a RKHS, then the approximation of the overall model is accurate point-wise everywhere via the same proof given in (Rosenfeld et al., 2022), and uniformly over compact sets when the RKHS consists of continuous functions. In contrast, the approximation is only accurate almost everywhere when considering Koopman operators posed over $L^2(\mathbb{R})$. Point-wise everywhere approximation is a distinctive advantage of kernel based methods. This convergence result is less clear from an invocation of the kernel trick in machine learning, and exemplifies the advantage of the operator theoretic considerations introduced in this manuscript.

The approximation of the Koopman operator in the operator norm topology naturally leads to the question of when a Koopman operator, or more generally, a composition operator, can be compact. Compactness is a central issue for DMD as every approximation is indeed of finite rank, stemming from the observed data. In 1979, it was established in (Singh & Kumar, 1979) that composition operators over $L^2(\mu)$ cannot be compact when $\mu$ is non-atomic. Indeed, $L^2(\mathbb{R})$ has no compact composition operators.

Koopman operators over most frequently considered RKHSs are compact for a very narrow range of dynamics. In fact, most Koopman operators over these spaces are not even bounded, as will be expanded upon in Section 4.2.3. Unboundedness is an added complication for the valid implementation of DMD, addressed in Section 5, using densely defined and potentially unbounded Koopman operators. Alternatively, other classes of compact operators over RKHSs connected to dynamical systems can be leveraged for DMD procedures to give convergence guarantees (see, e.g., Rosenfeld et al. (2022)).

The remainder of this section introduces densely defined Koopman operators over RKHSs, where Lemma 1 enables the DMD algorithm introduced in Section 5.

**Definition 4.** Let $H$ be a RKHS over $\mathbb{R}^n$. For a function $F : \mathbb{R}^n \to \mathbb{R}^n$ we define the Koopman Operator (sometimes called a composition operator), $\mathcal{K}_F : \mathcal{D}(\mathcal{K}_F) \to H$, as $\mathcal{K}_Fg = g \circ F$ where $\mathcal{D}(\mathcal{K}_F) = \{g \in H : g \circ F \in H\}$. When $\mathcal{D}(\mathcal{K}_F)$ is dense in $H$, $\mathcal{K}_F$ is said to be densely defined.

While not all densely defined Koopman operators over RKHSs are bounded, they are all closed operators (Pedersen, 2012).

**Lemma 1.** *Let $F : X \to X$ be the symbol for a Koopman operator over a RKHS $H$ over a set $X$. $\mathcal{K}_F : \mathcal{D}(\mathcal{K}_F) \to H$ is a closed operator.*

The proof of the above lemma is provided in Appendix B.1.

**Proposition 4.** *If a Koopman operator is densely defined, its adjoint is densely defined and closed.*

*Proof.* Given the kernel function centered at $x$,

$$\langle \mathcal{K}_F g, K_x \rangle_H = \langle g \circ F, K_x \rangle_H = g(F(x)) = \langle g, K_{F(x)} \rangle_H.$$

Therefore, the linear functional $g \mapsto \langle \mathcal{K}_F g, K_x \rangle$ is bounded over $H$. Thus, $K_x \in \mathcal{D}(\mathcal{K}_F^*)$ for all $x \in X$, and $\mathcal{K}_F^* K_x = K_{F(x)}$. Hence, each kernel function is in the domain of the adjoint of a densely defined Koopman operator, and as the span of kernel functions is dense in their RKHS, the adjoint is densely defined. $\square$

# 4 A Different Landscape for Koopman Operators

This section examines properties of Koopman operators over RKHSs. The selection of space fundamentally changes the behavior of Koopman operators over that space, where properties such as the lattice of eigenfunctions, common eigenfunctions for different discretizations, and boundedness of the operators may not hold. In the succeeding subsections we discuss each of these properties and provide counter examples for each of these properties in Appendix A.3 for Koopman operators corresponding to the continuous time dynamics $\dot{x} = \begin{pmatrix} x_2 & -x_1 \end{pmatrix}^T$. We discuss that many RKHSs only support bounded Koopman operators when the discretized dynamics are affine, and in the case of the Gaussian RBF's native space, we provide a novel proof of this fact in Appendix B.2.

Subsequently, the notation $F_{\Delta t}$ will denote the discretized dynamics corresponding to $\dot{x} = \begin{pmatrix} x_2 & -x_1 \end{pmatrix}^T$ with fixed time step $\Delta t > 0$.

Recently, kernel methods have been adapted for the study of DMD and Koopman operators, largely through the guise of extended DMD, where kernels are leveraged to simplify computations via the kernel trick. However, the adjustment from the classical study of Koopman operators through ergodic theory to that of reproducing kernel Hilbert spaces leads to significant differences in the Koopman operators and their properties. In most cases, the ergodic theorem cannot be directly applied to recover invariants of the operation $g \mapsto g \circ F$, for a given $F$, since those invariants may be nonsmooth. This section exemplifies some of the distinguishing properties of Koopman operators over RKHSs, and in some cases, illustrates their limitations.

Much of the classical properties of Koopman operators is established in a variety of specific contexts, such as $L^2$ spaces of invariant measures and $L^1$ spaces, can be seen in (Budišić et al., 2012; Kawahara, 2016; Kutz et al., 2016b; Brunton & Kutz, 2019; Brunton et al., 2021). Properties of the Koopman operator strongly depend on the selection of underlying vector space, and boundedness, compactness, eigenvalues, etc. change based on this selection. While Koopman operators were introduced by Koopman in 1931 in (Koopman, 1931) and then later picked up by the data science community in the early 2000s (e.g. (Mezić, 2005; Kutz et al., 2016b)), the study of such operators and their properties continued in earnest throughout the 20th century as composition operators (e.g. (Shapiro, 2012)). This is particularly important for RKHSs, where the specification of a bounded or densely defined Koopman operator over a particular space yields strong restrictions on the available dynamics.

## 4.1 Concerning Sampling and Discretizations

### 4.1.1 Forward Complete Dynamics

In applications, Koopman operators enter the theory of continuous time dynamics through a discretization of the continuous time dynamical system (Bittracher et al., 2015; Mauroy & Mezić, 2016). That is, given the dynamical system $\dot{x} = f(x)$, the system is discretized through the selection of a fixed time-step, $\Delta t > 0$, as $x_{m+1} = x_m + \int_{t_m}^{t_m + \Delta t} f(x(t))dt$, where the right hand side plays the role of the discrete dynamics. However,

for such a discretization to exist for arbitrarily large values of $m$, it is necessary that the dynamics be *forward complete*. The forward completeness assumption restricts the class of continuous dynamics on which Koopman based methods may be applied. For example, the continuous time dynamics, $\dot{x} = 1 + x^2$, does not admit a discretization, since it is not forward complete. Example 2 in Appendix A.2 demonstrates where discretization fails for $\dot{x} = 1 + x^2$. A DMD method that circumvents this requirement, by utilizing Liouville operators and occupation kernels, may be found in (Rosenfeld et al., 2022).

### 4.1.2  Sampling and Data Science

Ergodic based methods as employed in (Budišić et al., 2012; Mezić, 2005; Kutz et al., 2016b;b; Takeishi et al., 2017) provide a methodology for obtaining invariants and eigenfunctions for a Koopman operator almost everywhere. That is, by selecting a continuous representative of an equivalence class in an $L^2$ space for the invariant measure, at almost every point within the domain, time averaging against that representative will converge to an invariant of the operator. However, this is a "probability 1" result, and the number of points where it may fail can potentially be uncountable. Without any external information concerning the convergence, there is no true guarantee that at a particular selected point, the time-averaged approximation will be close to the evaluation of an actual invariant at that point. Such computational issues are precisely where the strength of kernel methods manifest. To illustrate the kernel method consider the following theorem:

**Theorem 1.** *Let $F : \mathbb{R}^n \to \mathbb{R}^n$ be a discretization of a dynamical system with the corresponding Koopman operator, $\mathcal{K}_F : H \to H$, and let $H$ be a RKHS over $\mathbb{R}^n$ consisting of continuous functions. Suppose further that $\epsilon > 0$ and $\hat{\mathcal{K}}_F : H \to H$ is an approximation of $\mathcal{K}_F$ such that the norm difference is bounded as $\|\mathcal{K}_F - \hat{\mathcal{K}}_F\| < \epsilon$. Suppose that $\hat{\varphi} \in H$ and is a normalized eigenfunction of $\hat{\mathcal{K}}_F$ with eigenvalue $\lambda$. Then $\hat{\varphi}$ is an approximate eigenfunciton of $\mathcal{K}_F$.*

*Proof.*

$$|\mathcal{K}_F\hat{\varphi}(x) - \lambda\hat{\varphi}(x)| = |\mathcal{K}_F\hat{\varphi}(x) - \hat{\mathcal{K}}_F(x)\hat{\varphi}(x)|$$
$$= |\langle(\mathcal{K}_F - \hat{\mathcal{K}}_F)\hat{\varphi}, K(\cdot, x)\rangle_H| \leq \|\mathcal{K}_F - \hat{\mathcal{K}}_F\|\|K(\cdot, x)\|_H \leq \epsilon \cdot C$$

and $C > 0$ is a constant that depends on the kernel function and a prespecified compact domain. The compact domain may be extended to all of $\mathbb{R}^n$ in some cases, such as when the kernel function is the Gaussian RBF kernel function. $\square$

Thus it can be seen that kernel spaces and approximations that are close to the Koopman operator, in operator norm, can provide functions that behave similar to eigenfunctions of the Koopman operator. Moreover, the difference in behavior from a proper eigenfunction is governed pointwise by how close the operator approximation is in the first place.

### 4.2  Properties of the Operators

This section will consider the classical Fock space consisting of entire functions as a function space over which the Koopman operator is defined. The Fock space is used extensively in Quantum Mechanics (Hall, 2013) and it is a space where operators have been well studied (Zhu, 2012).

**Definition 5.** The Fock space is a RKHS, with kernel function $K(z, w) = e^{\bar{w}z}$. The kernel function for the Fock space over $\mathbb{C}^n$ may be obtained through a product of single variable kernels as $K(z, w) = e^{w^*z} = e^{\bar{w}_1 z_1} \cdots e^{\bar{w}_n z_n}$. The Fock space is given as

$$F^2(\mathbb{C}) := \left\{ f(z) = \sum_{m=0}^{\infty} a_m z^m : \sum_{m=0}^{\infty} |a_m|^2 m! < \infty \right\}.$$

Closely related to the Fock space is the exponential dot product kernel, $e^{x^T y}$, where for a single variable, the exponential dot product kernel's native space may be obtained by restricting the Fock space to the reals, and

then taking the real part of the restricted functions. Through a conjugation of the exponential dot product kernel, the Gaussian RBF may be obtained as

$$K_G(x, y) = e^{-\|x\|_2^2/2} e^{x^T y} e^{-\|y\|_2^2/2} = \exp\left(-\frac{\|x - y\|_2^2}{2}\right),$$

and performing the same operation on the Fock space kernel over $\mathbb{C}^n$ yields

$$K_G(z, w) = e^{-z^2/2} e^{w^* z} e^{-\bar{w}^2/2} = \exp\left(-\frac{(z - \bar{w})^2}{2}\right),$$

which is the kernel corresponding to the complexified native space for the Gaussian radial basis function over $\mathbb{C}^n$ (cf. (Steinwart & Christmann, 2008)). This space may be expressed as

$$H_G^2(\mathbb{C}) = \left\{ g(z) e^{-z^2/2} : g \in F^2(\mathbb{C}^n) \right\},$$

and the native space corresponding to the Gaussian RBF can be obtained by taking the real parts of functions from $H_G^2$ and restricting to $\mathbb{R}^n$.

### 4.2.1 Lattice of Eigenfunctions

As presented in (Budišić et al., 2012; Klus et al., 2015), the eigenfunctions of Koopman operators over $L^\infty(\mathbb{R})$ form a lattice. That is if $\varphi_1$ and $\varphi_2$ are two eigenfunctions for the Koopman operator, then so is $\varphi_1 \cdot \varphi_2$. For the lattice to occur more generally, it is necessary for the product of the eigenfunctions to be a member of the underlying vector space. This closure property holds, for example, in the space of continuous functions and other Banach algebras. Hilbert spaces are not generally Banach algebras, and since it is desirable to work over Hilbert spaces for properties such as best approximations, projections, and orthonormal bases (cf. (Folland, 1999)), it is important to point out that the closure property of eigenfunctions of Koopman operators does not hold in general. For example the eigenfunctions of $\mathcal{K}_{F_\pi}$ do not form a lattice. Fundamentally, powers of $\varphi(z) = e^{z^2/4} \in F^2(\mathbb{C})$ are not all contained in the Fock space. This is ultimately a consequence of growth conditions imposed by the RKHS norm. For more details consult Appendix A.3.1.

### 4.2.2 Common Eigenfunctions

The intuition behind the use of Koopman operators in the study of continuous time dynamical systems is that eigenfunctions for the Koopman operators should be "close" to that of the Koopman generator for small timesteps. However, semi-groups of Koopman operators do not always share a common collection of eigenfunctions. Since each Koopman operator obtained through a fixed time-step may produce a different collection of eigenfunctions, there is no way to distinguish which, if any, should correspond to eigenfunctions of the Koopman generator. In Appendix A.3.2 we show that an eigenfunction for the Koopman operator corresponding to $F_{\pi/2}$ and is not an eigenfunction for the Koopman operator corresponding to $F_{\pi/3}$.

### 4.2.3 Boundedness of Koopman Operators

Throughout the literature, it is frequently assumed that Koopman operators are bounded. This assumption manifests as an unrestricted selection of observables in the study of the Koopman operator. When a Koopman operator is a densely defined operator whose domain is the entire Hilbert space, it is also closed (Pedersen, 2012). Hence, by the closed graph theorem (cf. (Folland, 1999, Theorem 5.12)), such an operator must be bounded. Furthermore, the collection of finite rank operators is dense in the collection of bounded operators over a Hilbert space in the strong operator topology (SOT) (cf. (Pedersen, 2012, Paragraph 4.6.2)). Convergence in SOT was independently studied in the work (Korda & Mezić, 2018), where the DMD routine was demonstrated to converge to a bounded Koopman operator in SOT.

As mentioned in (Korda & Mezić, 2018), SOT convergence does not in general lead to convergence of the eigenvalues. To preserve spectral convergence, the finite rank approximations produced by DMD algorithms need to converge to Koopman operators in the operator norm topology. The most direct approach, and

one that leads to good pointwise estimates of eigenfunctions, is through the use of compact Koopman operators. However, it isn't immediately clear when one can expect a continuous dynamical system to yield a compact Koopman operator through discretization. For example, the Koopman operator corresponding to discretization of the continuous time system $\dot{x} = 0$ is the identity operator, $I$, for any fixed time step, and $I$ is not compact over any infinite dimensional Hilbert space.

In addition, for any given RKHS, the collection of bounded Koopman operators is very small. It was demonstrated in (Carswell et al., 2003) that a Koopman operator over the Fock space is bounded only when the corresponding discrete dynamics are *affine*. It follows that the same result holds over the exponential dot product kernel's native space.

It may perhaps be less obvious that this result extends to the Gaussian RBF's native space. A proof of this fact was first published in (Ikeda et al., 2022). We provide a novel proof of this result that leverages the Gaussian RBF kernel's connection to the Fock space in Appendix B.2. The theorem and corollary related to our proof is shown below. These proofs demonstrate that even for popular selections of RKHSs, the collection of bounded Koopman operators is small.

**Theorem 2.** *If $F : \mathbb{C}^n \to \mathbb{C}^n$ is an entire function, and $\mathcal{K}_F$ is bounded on $H_G$, then $F(z) = Az + b$ for a matrix $A \in \mathbb{C}^{n \times n}$ and vector $b \in \mathbb{C}^n$. Here, $H_G$ represents the Gaussian RBF's native space.*

**Corollary 1.** *If $F$ is a real entire vector valued function, and $\mathcal{K}_F$ is bounded on the Gaussian RBF's native space over $\mathbb{R}^n$, then $F$ is affine.*

Hence, for the most commonly used kernel function in machine learning, the collection of bounded (and hence compact) Koopman operators over its native space is restricted to only those Koopman operators corresponding to affine dynamics. Each selection of RKHS and kernel function will yield a correspondingly small collection of bounded Koopman operators. It should be noted that Koopman operators were completely classified over the classical sampling space, the Paley-Wiener space (Chacón & Giménez, 2007), as also being those that correspond to affine dynamics, and it is a simple exercise to show that the native space for the polynomial kernel also only admits bounded Koopman operator when the dynamics are affine.

In summary, thus far it has been established that Koopman operators are only bounded in the case where the dynamics are affine over the following spaces:

- The polynomial kernel space, $K(x, y) = (1 + x^T y)^n$,

- The Fock Space, $K(z, w) = e^{\alpha z \bar{w}}$,

- The exponential dot product kernel's native space, $K(x, y) = e^{\alpha x^T y}$,

- The Gaussian RBF's native space, $K(x, y) = \exp\left(-\frac{1}{\mu}\|x - y\|_2^2\right)$,

- Paley-Wiener spaces, $K(x, y) = sinc(\alpha(x - y))$, and

- Other spaces discussed in (Ikeda et al., 2022).

Consequently, in most practical respects Koopman operators over RKHSs should not be assumed to be bounded, and certainly not compact.

# 5 Dynamic Mode Decomposition with Koopman Operators over RKHSs

As a product of its genesis in the machine learning community, many DMD procedures appeal to feature space, and this holds in implementations of kernel-based extended DMD (Williams et al., 2015b), which casts the snapshots from a finite dimensional nonlinear system into an infinite feature space. The direct involvement of the feature space in the estimation of the Koopman operator leads to rather complicated numerical machinery. To avoid directly computing the infinite dimensional vectors that result, an involved collection of linear algebra techniques are leveraged to extract the Koopman modes. Here it is shown that this process may be simplified and that a procedure that directly involves the kernel functions centered at

the snapshots simplifies the design of DMD algorithms. This approach keeps with the spirit of the "kernel trick," where feature vectors are never directly evaluated and only accessed through evaluations of the kernel function itself.

The algorithm presented in this section is designed around scalar valued RKHSs, which necessitates the decomposition of the (vector valued) full state observable component-wise. A complete vector valued algorithm is discussed in Section 6, where it is demonstrated that the present algorithm is computationally equivalent to the vector valued algorithm for certain selections of kernel operators.

Recall from Proposition 2 that in a real valued Hilbert space, the projection of a function $g$ onto a collection of linearly independent basis functions, $u_1, \ldots, u_M$, is a linear combination of those functions, $Pg = \sum_{j=1}^{M} w_j u_j$ where the weights $w_j$ may be determined by solving

$$
\begin{pmatrix} \langle u_1, u_1 \rangle_H & \cdots & \langle u_1, u_M \rangle_H \\ \vdots & \ddots & \vdots \\ \langle u_M, u_1 \rangle_H & \cdots & \langle u_M, u_M \rangle_H \end{pmatrix} \begin{pmatrix} w_1 \\ \vdots \\ w_M \end{pmatrix} = \begin{pmatrix} \langle g, u_1 \rangle_H \\ \vdots \\ \langle g, u_M \rangle_H \end{pmatrix}.
$$

We will use Definition 6 in order to aid us in creating and denoting the finite rank representation of $\mathcal{K}_F$.

**Definition 6.** Let $E$ be a linear transformation between two finite dimensional vector spaces $V$ and $W$. Suppose that $\alpha = \{\alpha_1, \ldots, \alpha_N\}$ is an ordered basis for $V$, and suppose that $\beta = \{\beta_1, \ldots, \beta_M\}$ is an ordered basis for $W$. A matrix representation of the linear transformation, $E$, with respect to the ordered bases $\alpha$ and $\beta$ is denoted as $[E]_\alpha^\beta$, where the $j$-th column of this matrix contains the weights of the vector $E_{\alpha_j}$ corresponding to the ordered basis $\beta$.

Throughout this algorithm, a Koopman operator will be assumed to be densely defined, as Section 4 demonstrated that most Koopman operators cannot be expected to be bounded or compact. An additional assumption will be made that the kernel functions themselves reside in the domain of the Koopman operator. It should be noted that since the kernels are always in the domain of the adjoint of the Koopman operator (see Section 3), a finite rank representation of the adjoint of the Koopman operator may thus be derived without assuming that the kernels are in the domain of the Koopman operator.

For the sake of the derivation, it is also assumed that the Koopman operator is diagonalizable, which is not generally expected to be true. However, the finite rank representations leveraged in this manuscript are almost always diagonalizable, since the set of non-diagonalizable matrices are of measure zero in the collection of all matrices. Moreover, for periodic data sets, the adjoint of the Koopman operator will be invariant on the span of the collection of kernel functions centered at the snapshots, and thus, the finite rank representations will be explicitly the adjoint of the Koopman operator on that subspace, which supports the assumption of the availability of eigendecompositions for the Koopman operator in the periodic or quasiperiodic settings.

For a given collection of snapshots $\{x_1, x_2, ..., x_m\}$[1], the goal is to determine a finite rank representation of $\mathcal{K}_F$ that is derived from the kernel functions centered at the snapshots. To express a finite rank representation, the ordered basis $\alpha = \{k_{x_1}, ..., k_{x_{m-1}}\}$ is leveraged. In particular, if $P_\alpha$ is the projection onto $\text{span}(\alpha)$, the operator $P_\alpha \mathcal{K}_F$ maps $\text{span}(\alpha)$ to itself, which enables the discussion of eigenfunctions and eigenvalues of $P_\alpha \mathcal{K}_F$ using only functions in $\text{span}(\alpha)$.

**Proposition 5.** *Given a collection of snapshots* $\{x_1, x_2, ..., x_m\} \subset X$ *and the corresponding ordered basis* $\alpha = \{k_{x_1}, ..., k_{x_{m-1}}\}$ *of kernel functions, if* $g = \sum_{i=1}^{m-1} a_i k_{x_i}$ *and* $P_\alpha \mathcal{K}_F g = \sum_{i=1}^{m-1} b_i k_{x_i}$ *then*

$$
Gb = \mathcal{I}a, \tag{2}
$$

---

[1]While availability of a time series of snapshots $\{x_1, x_2, ..., x_m\}$ such that $x_{i+1} = F(x_i)$ is a more typical use case, the developed method does not require such a time series. It can also be implemented using arbitrary snapshots $\{x_1, x_2, ..., x_m\}$ and $\{y_1, y_2, ..., y_m\}$ provided $y_i = F(x_i)$.

*where $a := (a_1, ..., a_{m-1})^T$, $b := (b_1, ..., b_{m-1})^T$, and the Gram matrix $G$ and the interaction matrix $\mathcal{I}$ are given by*

$$G := \begin{pmatrix} K(x_1, x_1) & \cdots & K(x_1, x_{m-1}) \\ \vdots & \ddots & \vdots \\ K(x_{m-1}, x_1) & \cdots & K(x_{m-1}, x_{m-1}) \end{pmatrix}, \quad and \quad \mathcal{I} := \begin{pmatrix} K(x_2, x_1) & \cdots & K(x_2, x_{m-1}) \\ \vdots & \ddots & \vdots \\ K(x_m, x_1) & \cdots & K(x_m, x_{m-1}) \end{pmatrix}.$$

*Proof.* Using Proposition 2, the reproducing property, and the fact that $\mathcal{K}_F k_{x_i}(x) = k_{x_i}(F(x)) = K(F(x), x_i)$,

$$Gb = \begin{pmatrix} \langle \mathcal{K}_F \sum_{i=1}^{m-1} a_i k_{x_i}, k_{x_1} \rangle_H \\ \vdots \\ \langle \mathcal{K}_F \sum_{i=1}^{m-1} a_i k_{x_i}, k_{x_{m-1}} \rangle_H \end{pmatrix} = \begin{pmatrix} \sum_{i=1}^{m-1} a_i \langle \mathcal{K}_F k_{x_i}, k_{x_1} \rangle_H \\ \vdots \\ \sum_{i=1}^{m-1} a_i \langle \mathcal{K}_F k_{x_i}, k_{x_{m-1}} \rangle_H \end{pmatrix} = \begin{pmatrix} \sum_{i=1}^{m-1} a_i K(x_2, x_i) \\ \vdots \\ \sum_{i=1}^{m-1} a_i K(x_m, x_i) \end{pmatrix} = \mathcal{I}a.$$

$\square$

If $k_{x_1}, \dots, k_{x_{M-1}}$ are linearly independent, then $G$ is invertible and $b = G^{-1}\mathcal{I}a$, i.e., the operator $P_\alpha \mathcal{K}_F$, restricted to span$(\alpha)$ is uniquely represented by the matrix $[P_\alpha \mathcal{K}_F]^\alpha_\alpha = G^{-1}\mathcal{I}$. If $G$ is singular, then the projected function $P_\alpha \mathcal{K}_f g$ admits multiple sets of coefficients $b_i$ that satisfy $P_\alpha \mathcal{K}_f g = \sum_{i=1}^{m-1} b_i k_{x_i}$. In this case, the finite rank representation is not unique. In this paper, we study two specific approaches to compute a finite rank representation for the case where $G$ is singular or numerically singular.

The first approach, summarized in Algorithm 1, is regularized regression, i.e., $[P_\alpha \mathcal{K}_F]^\alpha_\alpha := (G + \epsilon I_{m-1})^{-1}\mathcal{I}$, where $\epsilon > 0$ is a user-selected regularization coefficient and $I_{m-1}$ is an $m - 1 \times m - 1$ identity matrix.

The second approach is a truncated pseudoinverse approach, i.e., $[P_\alpha \mathcal{K}_F]^\alpha_\alpha := G^+_\epsilon \mathcal{I}$, where $G^+_\epsilon$ is the truncated pseudoinverse of $G$, computed by zeroing the singular values of $G$ that are smaller than $\epsilon \geq 0$.

It should be noted that $G$ and $\mathcal{I}$ are precisely the matrices examined in (Williams et al., 2015b) after the use of a truncated SVD. If the truncated pseudoinverse approach is used to implement the method developed in this paper, then the generated results are identical to those generated by the KDMD method in (Williams et al., 2015b). The results obtained by using the regularized regression approach differ from those obtained using the KDMD method in (Williams et al., 2015b), but not substantially so.

In Appendix C we provide empirical evidence for the similarity of the results obtained from our method and the method employed in (Williams et al., 2015b).

The objective of DMD is to use the finite rank representation determined above to create a data driven model of the dynamical system. This makes use of a fundamental property of eigenfunctions of the Koopman operator.

**Lemma 2.** *Suppose that $\varphi$ is an eigenfunction of $\mathcal{K}_F$ with eigenvalue $\lambda$. Evaluating the eigenfunction at a snapshot reveals $\varphi(x_{i+1}) = \lambda^i \varphi(x_1)$.*

*Proof.* Let $\varphi$ be an eigenfunction of $\mathcal{K}_F$ with eigenvalue $\lambda$, where $F$ represents the discrete time dynamics such that $x_{i+1} = F(x_i)$. Furthermore, recall the definition of the Koopman operator presented in Definition 4. Then

$$\lambda\varphi(x_i) = \mathcal{K}_F\varphi(x_i) = \varphi(F(x_i)) = \varphi(x_{i+1})$$

Therefore, $\varphi(x_{i+1}) = \lambda^i \varphi(x_1)$. $\square$

**Proposition 6.** *Suppose that $\{\varphi_j\}_{j=1}^\infty$ is a complete set of eigenvectors of the Koopman operator, $\mathcal{K}_F$, corresponding to the eigenvalues $\{\lambda_j\}_{j=1}^\infty$. For a state $x \in \mathbb{R}^n$, let $(x)_d$ be the $d$-th component of $x$ for $d = 1, \dots, n$. If it is assumed that the mapping $x \mapsto (x)_d$ is in the RKHS (as it is when $H$ is the native space for the exponential dot product space (Steinwart & Christmann, 2008)), then each snapshot may be reconstructed as*

$$x_{i+1} = \lim_{M \to \infty} \sum_{j=1}^M \xi_{j,M} \lambda_j^i \varphi_j(x_1). \tag{3}$$

*Proof.* Since $\{\varphi_j\}_{j=1}^{\infty}$ is an eigenbasis for $\mathcal{K}_F$ corresponding to the eigenvalues $\{\lambda_j\}_{j=1}^{\infty}$ and the mapping $x \mapsto (x)_d$ is in the RKHS,

$$(x)_d = \lim_{M\to\infty} \sum_{j=1}^{M} (\xi_{j,M})_d \varphi_j(x)$$

for some coefficients $\{(\xi_{j,M})_d\}_{j=1}^{\infty}$. By stacking each $(x)_d$, the full state observable $g_{\mathrm{id}}$, given by $g_{\mathrm{id}}(x) = x$, is expressed as

$$g_{\mathrm{id}}(x) = \lim_{M\to\infty} \sum_{j=1}^{M} \xi_{j,M} \varphi_j(x). \tag{4}$$

Therefore, equation 3 can be derived by using equation 4 and Lemma 2. □

Note that since the Koopman operator is not generally a normal operator, $\{\varphi_i\}_{i=1}^{\infty}$ is not expected to be an orthonormal basis, and hence, there may be nonzero influences between the coefficients obtained by projection and this is expressed by the additional index $M$ in $\xi_{j,M}$. This means that a series representation of the decomposition as expressed in (Kawahara, 2016; Brunton & Kutz, 2019) is not always possible. *Hence, Koopman modes are not fixed quantities unless there is an orthonormal basis of eigenfunctions for the Koopman operator.*

Since the Koopman operator is approximated here by a finite rank representation, perfect reproduction of $g_{\mathrm{id}}$ through a series of eigenfunctions is not possible. Instead, eigenfunctions determined through the finite rank representation are used to construct the approximation of $g_{\mathrm{id}}$. In particular, the matrix $[P_\alpha \mathcal{K}_F]_\alpha^\alpha$ is the matrix representation of $P_\alpha \mathcal{K}_F$.

**Proposition 7.** *If $v_j$ is an eigenvector for the matrix $[P_\alpha \mathcal{K}_F]_\alpha^\alpha$ with eigenvalue $\lambda_j$, then $\sum_{i=1}^{m-1} (v_j)_i K(x, x_i)$ is an eigenfunction of $P_\alpha \mathcal{K}_F$.*

*Proof.* By the definition of eigenvector, $[P_\alpha \mathcal{K}_F]_\alpha^\alpha v_j = \lambda_j v_j$. Therefore,

$$P_\alpha \mathcal{K}_F \left( \sum_{i=1}^{m-1} (v_j)_i K(x, x_i) \right) = \begin{pmatrix} K(x, x_1) \\ \vdots \\ K(x, x_{m-1}) \end{pmatrix}^T [P_\alpha \mathcal{K}_F]_\alpha^\alpha v_j = \lambda_j \sum_{i=1}^{m-1} (v_j)_i K(x, x_i).$$

□

Using Proposition 7 the corresponding normalized eigenfunction is denoted by

$$\hat{\varphi}_j(x) := \frac{1}{\sqrt{v_j^\dagger G v_j}} \sum_{i=1}^{m-1} (v_j)_i K(x, x_i), \tag{5}$$

where $G = (K(x_i, x_\ell))_{i,\ell=1}^{m-1}$ is the Gram matrix associated with the snapshots and the kernel function and $(\cdot)^\dagger$ denotes the conjugate transpose.

Using a finite rank representation of equation 4, it is easy to see that the $d$-th row of the matrix $\hat{\xi} := \begin{pmatrix} \hat{\xi}_1 & \dots & \hat{\xi}_{m-1} \end{pmatrix}$ of Koopman modes is comprised of the components of $(x)_d$ along the (non-orthogonal) directions $\hat{\varphi}_j$. That is,

$$g_{\mathrm{id}}(x_i) = x_i = \sum_{j=1}^{m-1} \xi_j \hat{\varphi}_j(x_i), \tag{6}$$

which yields

$$\hat{\xi} V^T G = X,$$

where $X := \begin{pmatrix} x_1 & \dots & x_{m-1} \end{pmatrix}$ is the data matrix and

$$V := \left( \frac{v_1}{\sqrt{v_1^\dagger G v_1}} \quad \cdots \quad \frac{v_{m-1}}{\sqrt{v_{m-1}^\dagger G v_{m-1}}} \right)$$

is the matrix of normalized eigenvectors of $[P_\alpha \mathcal{K}_F]_\alpha^\alpha$.

Similar to the finite rank representation above, if $G$ is non-singular, then the matrix of Koopman modes is unique, given by $\hat{\xi} = X(V^T G)^{-1}$. If $G$ is singular and if regularized regression is used, then modes are computed using $\hat{\xi} = X(V^T(G + \epsilon I_{m-1}))^{-1}$. If the truncated pseudoinverse approach is selected, then eigenvalues of the finite rank representation with absolute value smaller than $\epsilon$ are removed from the computation, along with the corresponding eigenvectors, and the modes are computed using $\hat{\xi} = X(V^T G)_\epsilon^+$.

Using the approximate eigenfunctions, $\hat{\varphi}_j$, equation 6, and Lemma 2 a data driven model for the system is obtained as

$$x_{i+1} \approx \sum_{j=1}^{m-1} \hat{\xi}_j \lambda_j^i \hat{\varphi}_j(x_1). \tag{7}$$

Furthermore, the discrete-time dynamics $F$ may be approximated (by setting $i = 1$) as

$$F(x) \approx \hat{F}(x) := \sum_{j=1}^{m-1} \hat{\xi}_j \lambda_j \hat{\varphi}_j(x). \tag{8}$$

Consider a continuous-time system $\dot{x} = f(x)$ with $f \in \mathcal{C}^1$, and let a compact set $X$ be forward invariant under the flow of the system. If the discrete-time system $x_{k+1} = F(x_k)$ is obtained via discretization of the continuous-time system with time step $T$, and if $(e^{\gamma T}, \varphi)$ is an eigenpair of resulting Koopman operator $\mathcal{K}_T$ for all $T > 0$, then along a solution $x(\cdot)$ of $\dot{x} = f(x)$, we have $\frac{d\varphi(x(t))}{dt} = \gamma \varphi(x(t))$ (Mauroy & Mezić, 2016, Section III-A). Motivated by this relationship, the continuous-time dynamics may be approximated as

$$f(x) \approx \hat{f}(x) := \sum_{j=1}^{m-1} \frac{\log(\lambda_j)}{T} \hat{\xi}_j \hat{\varphi}_j(x). \tag{9}$$

Note that in the above construction, the eigenfunctions $\varphi_j$ are assumed to be common across all time-steps $T$. In view of Example 5, not all eigenfunctions of the Koopman operator may be common eigenfunctions. As a result, the continuous-time model in equation 9 is a heuristic, albeit a useful one (see Figure 3 in the appendix).

## 6    Vector Valued Considerations

The attentive reader will notice that the ultimate objective of DMD is to achieve an approximation or decomposition of the function $g_{\mathrm{id}}(x) := x$, the full state observable. However, the full state observable is $\mathbb{R}^n$ valued, whereas the RKHSs in question consist of scalar valued functions. Consequently, the coefficients that are determined to approximate the full state observable are vector valued. Up to now, the methods have simply separated the individual components of $g_{\mathrm{id}}$ and established an approximation for each of them separately. The weights for these approximations are stacked to make the vector valued coefficients.

This section shows how the vector valued coefficients arise naturally from a projection onto vector valued functions in a vector valued RKHS. With the right selection of kernel operator, this projection operation reduces to the setting of Section 5.

If the goal is to decompose the full state observable all at once, then one must appeal to vector valued RKHSs, which produce an operator valued kernel function.

**Proposition 8.** *Given a vector valued RKHS, $H$, consisting of functions that map a set $X$ to a Hilbert space $\mathcal{Y}$, there is for each $x \in X$ and $\nu \in \mathcal{Y}$ a function $K_{x,\nu} \in H$ such that $\langle g, K_{x,\nu} \rangle_H = \langle g(x), \nu \rangle_{\mathcal{Y}}$.*

---

**Algorithm 1:** Pseudocode for the kernel perspective based DMD algorithm. Upon obtaining the Koopman modes, the approximate eigenfunctions, and the eigenvalues, equation 7 is used to compute $x_{i+1}$.

---

**Input:** Snapshots, $X = \{x_1, x_2, ..., x_m\}$
**Input:** Kernel function $K : \mathbb{R}^n \times \mathbb{R}^n \to \mathbb{R}$ of an RKHS
  Compute the Gram matrix, $G = (K(x_i, x_\ell))_{i,\ell=1}^{m-1}$
  **if** $G$ is ill-conditioned **then**
    **if** regularized regression is used **then**
      Set $G = G + \epsilon I_{m-1}$
    **else if** truncated pseudoinverse is used **then**
      Set $G^{-1} = G_\epsilon^+$
    **end if**
  **end if**
  Compute the interaction matrix, $\mathcal{I} = (K(x_i, x_\ell))_{i=2,\ell=1}^{i=m,\ell=m-1}$
  Compute the finite rank representation of the Koopman operator, $[P_\alpha \mathcal{K}_F]_\alpha^\alpha = G^{-1}\mathcal{I}$
  Compute eigenvalues, $\lambda_j$, and eigenvectors, $v_j$, of $[P_\alpha \mathcal{K}_F]_\alpha^\alpha$
  **if** truncated pseudoinverse is used **then**
    For all $j$, if $|\lambda_j| < \epsilon$ then set $\lambda_j = 0$ and $v_j = 0$
    Set $(V^T G)^{-1} = (V^T G)_\epsilon^+$
  **end if**
  Compute the matrix of Koopman modes, $\hat{\xi} = X (V^T G)^{-1}$
  Compute approximate eigenfunctions, equation 5
**Output:** Koopman modes, $\hat{\xi}_j$ for $j = 1, \ldots, m-1$
**Output:** Approximate eigenfunctions, $\hat{\varphi}_j(x)$ for $j = 1, \ldots, m-1$
**Output:** Eigenvalues $\lambda_j$ for $j = 1, \ldots, m-1$

---

The mapping $\nu \mapsto K_{x,\nu}$ is linear, and hence, the kernels are expressed as $K_x\nu$ where $K_x : \mathcal{Y} \to H$ is a bounded operator. In this case, for each $g \in H$, $K_x^* g = g(x)$. The operator valued kernel associated with $H$ is given as

$$K(x, y) := K_x^* K_y : \mathcal{Y} \times \mathcal{Y} \to H,$$

and note that when $\mathcal{Y} = \mathbb{R}^n$, this operator valued kernel is a matrix whose entries are scalar valued functions.

When a vector valued RKHS, $H$, is obtained from a scalar valued RKHS, $\tilde{H}$ with kernel $k(x, y)$, as $H = \tilde{H}^n$ where the inner product of two elements, $g = (g_1, \ldots, g_n)^T$ and $h = (h_1, \ldots, h_n)^T$, from $H$ is given as $\langle g, h \rangle_H = \sum_{j=1}^n \langle g_j, h_j \rangle_{\tilde{H}}$, then the operator valued kernel is given as $k(x, y)I_n$, which has many convenient properties for computation. More details concerning vector valued RKHSs may be found in (Carmeli et al., 2010; Agler & McCarthy, 2023).

Here the Koopman operator is defined just as it would be for a scalar valued space: $\mathcal{K}_F g = g \circ F$, and the difference here is that $g$ is vector valued. Similar relationships between the Koopman operator and the kernel functions still hold, where

$$\langle \mathcal{K}_F g, K_x\nu \rangle_H = \langle g(F(x)), \nu \rangle_\mathcal{Y} = \langle g, K_{F(x)}\nu \rangle_H,$$

hence $\mathcal{K}_F^* K_x = K_{F(x)}$.

Now given a collection of snapshots, $\{x_1, \ldots, x_M\}$, the associated operator valued kernels are $K_{x_1}, \ldots K_{x_M}$. For decomposition of the full state observable, $g_{id}(x) = x$, one needs to interface with the Hilbert space and construct a finite rank approximation of $\mathcal{K}_F$ as was done in the prior section. This means that a subspace within $H$ needs to be constructed, say spanned by a collection of basis functions $\alpha$, so one can write $\tilde{\mathcal{K}}_F = P_\alpha \mathcal{K}_F P_\alpha$.

The selection of the basis $\alpha$ is less constrained than in the scalar valued case, where only the kernels centered at the snapshots were considered. In this case, not only must the snapshots be considered, but also complete

freedom of the choice of $\nu \in \mathbb{R}^n$ to select each $K_{x,\nu} = K_x \nu$. A natural choice is to select $\nu$ from the standard basis of $\mathbb{R}^n$. This leads to

$$\alpha = \{K_{x_0,e_1}, \ldots, K_{x_{M-1},e_1}, K_{x_0,e_2}, \ldots, K_{x_{M-1},e_2}, \ldots, K_{x_0,e_n}, \ldots, K_{x_{M-1},e_n}\},$$

which is a large basis for problems such as the cylinder flow example, where $n$ is of the order $80,000$. Consequently, the gram matrix corresponding to this basis is a square matrix of dimension $M \cdot n$.

So for arbitrary operator valued kernels, the DMD algorithm presented in this paper would have to invert an $M \cdot n$ dimensional matrix, and the computation time would scale with the dimension of the space as $O((M \cdot n)^3)$ using standard inversion algorithms. This scaling would make benchmarks such as the cylinder flow example completely infeasible.

Computational feasibility can be achieved through the judicious selection of the kernel operator. Here we leveraged a kernel operator of the form

$$K(x,y) = k(x,y)I_n,$$

with $k$ a scalar valued kernel function. In this case,

$$\langle K_x\nu, K_y\omega \rangle_H = k(x,y)\langle \nu, \omega \rangle_{\mathbb{R}^n}$$

and the two functions $K_x\nu$ and $K_y\omega$ are orthogonal when $\nu$ and $\omega$ are orthogonal in $\mathbb{R}^n$. This means that the Gram matrix composed of the $\alpha$ given above is going to be a block diagonal matrix with each block being the Gram matrix corresponding to the scalar valued kernel. Thus only one matrix of size $M$ must be inverted, and each dimension treated individually. Significantly, the inversion problem no longer scales with the dimension.

Alternatively, if we instead use a different scalar valued space for each dimension, we would get $n$ different Gram matrices to invert. The complexity would scale with the dimension, but linearly rather than cubicly.

**Proposition 9.** *Suppose that $k(x,y)$ corresponds to a scalar valued RKHS, $\tilde{H}$, and let $H$ be an $\mathbb{R}^n$ valued RKHS such that if $g \in H$, then $g = (g_1, \ldots, g_n)^T$ where $g_i \in \tilde{H}$ for each $i = 1, \ldots, n$ equipped with the inner product $\langle g, h \rangle_H = \sum_{i=1}^n \langle g_i, h_i \rangle_{\tilde{H}}$. Then $H$ is a vector valued RKHS.*

*Proof.* Let $v \in \mathbb{R}^n$ and $x \in X$ then

$$|\langle g(x), v \rangle_{\mathbb{R}^n}| \leq \|g(x)\|_{\mathbb{R}^n} \|v\|_{\mathbb{R}^n} = \sqrt{\sum_{i=1}^n |\langle g_i, k(\cdot, x) \rangle_{\tilde{H}}|^2} \|v\|_{\mathbb{R}^n}$$

$$\leq \sqrt{k(x,x)} \sqrt{\sum_{i=1}^n \|g_i\|_{\tilde{H}}^2} \|v\|_{\mathbb{R}^n} = \sqrt{k(x,x)} \|g\|_H \|v\|_{\mathbb{R}^n},$$

hence the functional $g \mapsto \langle g(x), v \rangle_{\mathbb{R}^n}$ is bounded. In this setting, $K(x,y) = \text{diag}(k(x,y), \ldots, k(x,y))$, and $\langle K_x e_i, K_y e_j \rangle_{\mathbb{R}^n} = 0$ if $i \neq j$. That $H$ is a vector valued RKHS follows from the above. $\square$

If $\{x_1, \ldots, x_N\}$ is a collection of snapshots such that $F(x_i) = x_{i+1}$, then a finite rank approximation of $\mathcal{K}_F$ may be constructed by examining the image of the operator on $K_{x_i,e_j}$ for $i = 1, \ldots, N$ and $j = 1, \ldots, n$, and then projecting back onto the span of these kernels. The projection operation requires the computation of the Gram matrix for the basis $\{K_{x_i,e_j}\}$, which is a block diagonal matrix, where each block corresponds to a selection of dimension through $e_j$. Thus, if $\tilde{G} = (k(x_s, x_\ell))_{s,\ell=1}^N$ the gram matrix manifests as

$$G = \begin{pmatrix} \tilde{G} & & \\ & \ddots & \\ & & \tilde{G} \end{pmatrix}.$$

Similarly, the interaction matrix is a block diagonal. Consequently, the vector of weights corresponding to each kernel function is composed of $n$ smaller vectors of length $N-1$, each one corresponding to a different dimension. Hence, each dimension may be treated independently. The eigenfunctions for this finite rank representation of the Koopman operator are then composed of $n$ identical collections of $N-1$ functions, differing only in the dimension they are supported in. Let $\tilde{\varphi}_1, \ldots, \tilde{\varphi}_N \in \tilde{H}$ be these scalar valued functions, and write $\varphi_{s,i} := \tilde{\varphi}_s e_i$ be the corresponding vector valued eigenfunction.

The full state observable is projected onto this collection of eigenfunctions as

$$g_{\mathrm{id}}(x) \approx \sum_{i=1}^{n} \sum_{s=1}^{N} w_{s,i} \varphi_{s,i}(x) = \sum_{s=1}^{N} \tilde{\varphi}_s(x) \left( \sum_{i=1}^{n} w_{s,i} e_i \right).$$

Here, $\sum_{i=1}^{n} w_{s,i} e_i = \xi_s$, where $\xi_s$ is the Koopman mode from the previous section.

## 7 Numerical Example

In this section we compare the results obtained from the developed DMD method and the method developed in (Williams et al., 2015b) on a benchmark dataset. Two additional numerical examples are included in Appendix C

### 7.1 Periodic flow around a cylinder

The website (Kutz et al., 2016a) accompanying the textbook (Kutz et al., 2016b) provides several data sets that serve as benchmarks for spectral decomposition approaches to nonlinear modeling, which have been released to the public through their website. This section utilizes the cylinder flow data set to demonstrate the results of the developed DMD method. The cylinder flow example is numerically generated, and the data provided corresponds to a planar steady state flow of the system. The data set consists of 151 snapshots, containing values of the vorticity of the flow at several mesh points in a plane, recorded every 0.02 seconds. In this demonstration, snapshots 1 through 30 are used as the input data, and snapshots 2 through 31 are used as output data, assuming that the $i$th and $(i+1)$th snapshots satisfy $x_{i+1} = F(x_i)$ for some unknown nonlinear function $F$. The snapshots are normalized so that the largest 2-norm among all snapshots is 1.

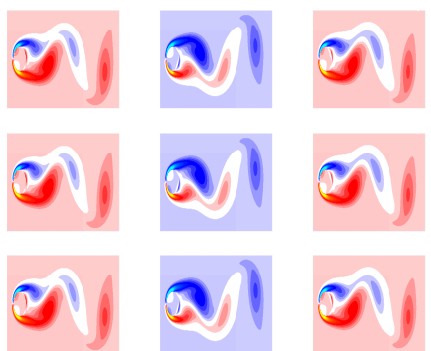

(a) This figure shows reconstruction of the vorticity field at the same time points using two different kernels. The left column shows reconstruction of the initial state, $x_1$. The middle column shows reconstructions of the state, $x_{15}$, and the right column corresponds to reconstruction of the state $x_{30}$.

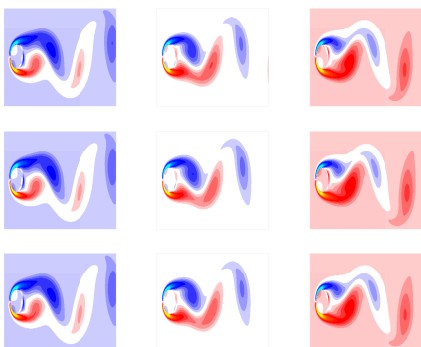

(b) This figure presents prediction of the vorticity field at the same time points using two different kernels. The left column presents prediction of the state $x_{51}$. The middle column shows prediction of the state $x_{101}$, and the right column corresponds to prediction of the state $x_{151}$.

Figure 1: Reconstruction and prediction of the vorticity of a fluid flow past a cylinder using two different kernel functions. The first rows contains the ground truth, the second rows leverages the Gaussian RBF kernel, and the third row uses the exponential dot product kernel.

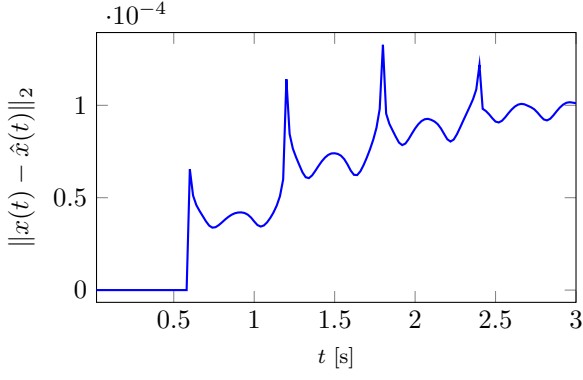
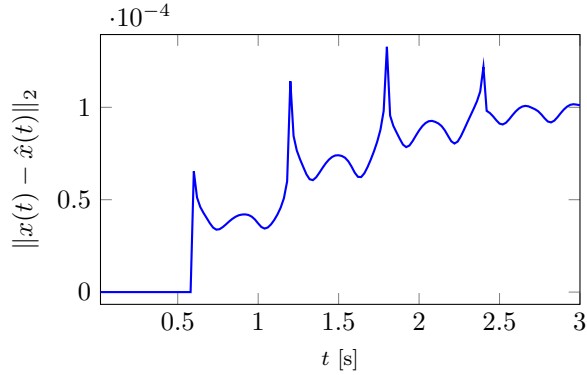

(a) Method developed in this paper

(b) Kernel DMD method in (Williams et al., 2015b)

Figure 2: For the periodic flow example, this figure shows the 2-norm of the error between the true snapshots and the snapshots generated using DMD. The results for the Gaussian RBF kernel and the exponential dot product kernel are identical.

The Koopman modes, eigenvalues, and eigenfunctions are then computed using the developed technique and snapshots 2 through 31 are reconstructed using equation 7. DMD is implemented using the exponential dot product kernel, $K(x,y) = \exp(\frac{1}{\mu}x^T y)$ (with $\mu = 1$), and the Gaussian RBF kernel, $K(x,y) = \exp\left(-\frac{1}{\mu}\|x - y\|_2^2\right)$ (with $\mu = 1$).

To further demonstrate the accuracy of the obtained finite-dimensional representation of the Koopman operator, snapshots 32 through 151 are *predicted* from snapshot 1 using equation 7. The code used to generate these results is publicly available, see Gonzalez et al. (2023).

## 7.2 Discussion

Figures 1a and 2 show the ability of the developed DMD technique to reconstruct the training data. The relative reconstruction errors associated with this system are of the order of $1e - 8$. Additionally, figure 1a shows that for the periodic flow example, similar reconstruction results can be obtained when using the Gaussian RBF kernel or the exponential dot product kernel.

As shown in figures 1b and 2, given the first 31 snapshots, the developed DMD technique is able to predict the remaining 120 snapshots, with a relative prediction error of order $1e - 4$, *without the knowledge of the underlying physics, F*. Furthermore, as expected, the performance of the developed method is nearly identical to the baseline Kernel DMD method developed in (Williams et al., 2015b).

## 8 Conclusion

This manuscript revisits theoretical assumptions concerning DMD of Koopman operators, including the existence of lattices of eigenfunctions, common eigenfunctions between Koopman operators, and boundedness and compactness of Koopman operators. Counterexamples that illustrate restrictiveness of the assumptions are provided for each of the assumptions. In particular, a major theoretical result is established to show that the native RKHS of the Gaussian RBF kernel function only supports bounded Koopman operators if the dynamics are affine. Moreover, a kernel-based DMD algorithm that simplifies the algorithm in (Williams et al., 2015a) and presents it in an operator theoretic context is developed and then validated through simulations. The developed DMD algorithm is also extended to vector valued RKHSs.

**Acknowledgments**

This research was supported by the Air Force Office of Scientific Research (AFOSR) under contract numbers FA9550-20-1-0127 and FA9550-21-1-0134, and the National Science Foundation (NSF) under award numbers

2027976, 2027999, and 1900364. Any opinions, findings and conclusions or recommendations expressed in this material are those of the author(s) and do not necessarily reflect the views of the sponsoring agencies.

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

## A    Examples

In this section we provide detailed examples that support some of the claims made throughout this paper. The title of each subsection is used as a way of linking the section in the paper with the examples that are relevant to that section.

### A.1    Introduction

**Example 1.** Given a Koopman operator $\mathcal{K}_F$ corresponding to the discrete dynamics, $F$, an $\epsilon > 0$, and a compact subset $\Omega$ that is invariant for the dynamics, it follows that if $\phi$ satisfies

$$\|K_F\phi - \lambda\phi\|_H \leq \epsilon,$$

then

$$|\phi(F^m(x)) - \lambda^m \phi(x)| \le C\epsilon \frac{1 - \lambda^{m+1}}{1 - \lambda}$$

for $x \in \Omega$, where the accuracy of the resultant models depend on $\epsilon$ and $\lambda$. Significantly, as $\epsilon$ tends to zero, so does the difference $|\phi(F^m(x)) - \lambda^m \phi(x)|$ at every point in $x \in \Omega$.

## A.2 Forward Complete Dynamics

**Example 2.** Consider the one dimensional dynamics:

$$\dot{x} = 1 + x^2.$$

For fixed $0 < \Delta t < \pi/2$, the corresponding discrete time dynamics are given as:

$$x_{m+1} = \tan(\arctan(x_m) + \Delta t).$$

Setting

$$x_m = \tan(\pi/2 - \Delta t),$$

it is clear that $x_{m+1}$ is undefined. Consequently, the composition symbol,

$$F(x) = \tan(\arctan(x) + \Delta t)$$

for the hypothetical Koopman operator, is not well defined over $\mathbb{R}^n$ for any selection of $\Delta t$.

## A.3 Properties of the Operators

The following simple example will be leveraged throughout the discussions in the ensuing subsections.

**Example 3.** Consider the dynamical system

$$\dot{x} = \begin{pmatrix} x_2 & -x_1 \end{pmatrix}^T,$$

which corresponds to circular dynamics in the plane. For any fixed $\theta := \Delta t$, the discretization of this system yields the linear discrete dynamics

$$x_{m+1} = \begin{pmatrix} \cos(\theta) & -\sin(\theta) \\ \sin(\theta) & \cos(\theta) \end{pmatrix} x_m.$$

That is, the discretization corresponding to a fixed time-step results in a fixed rotation of $\mathbb{R}^2$. To simplify the presentation, we use $\mathbb{C}$ as a model for $\mathbb{R}^2$, where rotation of the complex plane reduces to multiplication by a unimodular constant,

$$z_{m+1} = e^{i\theta} z_m. \tag{10}$$

The corresponding discrete time dynamics will be written as

$$F_\theta(z) := e^{i\theta} z.$$

### A.3.1 Lattice of Eigenfunctions

**Example 4.** Consider the dynamical system mentioned in example 3. If $\theta = \pi$ in equation 10, the discrete dynamics corresponding to rotation by $\pi$ in the complex plane become

$$z_{m+1} = e^{i\pi} z_m = -z_m.$$

The corresponding Koopman operator, $\mathcal{K}_{F_\pi} : F^2(\mathbb{C}) \to F^2(\mathbb{C})$, is given as

$$\mathcal{K}_{F_\pi} g(z) := g(-z).$$

Hence, every even function is an eigenfunction for this Koopman operator with eigenvalue 1.

Any function $g \in F^2(\mathbb{C})$ exhibits a strict bound on its growth rate (cf. (Zhu, 2012)). To wit,

$$|g(z)| = |\langle g, K(\cdot, z)\rangle_{F^2(\mathbb{C})}| \leq \|g\|_{F^2(\mathbb{C})}\|K(\cdot, z)\|_{F^2(\mathbb{C})} = \|g\|_{F^2(\mathbb{C})} e^{\frac{|z|^2}{2}}.$$

That is, if a function is in the Fock space then the function is of order at most 2, and if the function is of order 2 it has type at most $1/2$ (cf. (Boas, 2011)). Conversely, if an entire function is of order less than 2, it is in the Fock space, and if it is of order 2 and type less than $1/2$, then it is also in the Fock space. While functions of order 2 and type $1/2$ can be in the Fock space, it does not hold for every such function. For example, $e^{z^2/2}$ is of order 2 and type $1/2$, but is not in the Fock space.

Thus, $\varphi(z) = e^{z^2/4}$ is an eigenfunction for $\mathcal{K}_{F_\pi}$ in the Fock space. However, $\varphi \cdot \varphi = e^{z^2/2}$ is not in the Fock space, and cannot be an eigenfunction for $\mathcal{K}_{F_\pi} : F^2(\mathbb{C}) \to F^2(\mathbb{C})$. *Hence, the eigenfunctions for $\mathcal{K}_{F_\pi}$ do not form a lattice.*

### A.3.2 Common Eigenfunctions

**Example 5.** Consider again the dynamical system in example 3, but instead set $\theta = \pi/2$, then

$$F_{\pi/2}(z) = iz.$$

In this case, the polynomial $z^4 + z^8 \in F^2(\mathbb{C})$ is an eigenfunction for the Koopman operator corresponding to $\theta = \pi/2$ with eigenvalue 1. However, $z^4 + z^8$ is not an eigenfunction for $\mathcal{K}_{F_{\pi/3}}$, as

$$(e^{i\pi/3}z)^4 + (e^{i\pi/3}z)^8 = e^{i4\pi/3}z^4 + e^{i2\pi/3}z^8,$$

and the constants cannot be factored out of the polynomial as an eigenvalue.

## B  Proofs

### B.1  Proof of Lemma 1

**Lemma 1.** *Let $F : X \to X$ be the symbol for a Koopman operator over a RKHS $H$ over a set $X$. $\mathcal{K}_F : \mathcal{D}(\mathcal{K}_F) \to H$ is a closed operator.*

*Proof.* Suppose that $\{g_m\}_{m=1}^\infty \subset \mathcal{D}(\mathcal{K}_F)$ such that $g_m \to g \in H$ and $\mathcal{K}_F g_m \to h \in H$. To show that $\mathcal{K}_F$ is closed, we must show that $g \in \mathcal{D}(\mathcal{K}_F)$ and $\mathcal{K}_F g = h$. This amounts to showing that $h = g \circ F$, by the definition of $\mathcal{D}(\mathcal{K}_F)$. Fix $x \in X$, then

$$h(x) = \langle h, K_x\rangle_H = \lim_{m\to\infty} \langle \mathcal{K}_F g_m, K_x\rangle_H = \lim_{m\to\infty} g_m(F(x))$$
$$= \lim_{m\to\infty} \langle g_m, K_{F(x)}\rangle_H = \langle g, K_{F(x)}\rangle_H = g(F(x)).$$

As $x$ was an arbitrary point in $X$, $h = g(F(x))$ and the proof is complete. $\qquad\square$

### B.2  Proof Regarding Boundedness of the Koopman Operator

As stated in Section 4.2.3, here we provide a novel proof that shows that a Koopman operator over the Gaussian RBF's native space is bounded only when the corresponding discrete dynamics are affine. In order to achieve this we begin by introducing and proving a few lemmas.

**Lemma 4.** *If $\mathcal{K}_F$ is a bounded operator over the Gaussian RBF's native space, then $F$ is a real analytic vector valued function over $\mathbb{R}^n$.*

*Proof.* If $\mathcal{K}_F$ is bounded, then $\mathcal{K}_F K_y(x) = K_y(F(x)) = \exp(-\|F(x) - y\|_2^2)$ is in the RBF's native space for each $y \in \mathbb{R}^n$. Since every function in the RBF's native space is real analytic, so is $K_y(F(x))$, and thus, the logarithm, $-\|F(x) - y\|_2^2 = -\|F(x)\|_2^2 + 2y^T F(x) - \|y\|_2^2$ is real analytic. This holds if $y = 0$, and hence $\|F(x)\|_2^2$ is real analytic. Thus, for every $y$, the quantity $y^T F(x)$ is real analytic. That each component of $F(x)$ is real analytic follows from the selection of $y$ as the cardinal basis elements of $\mathbb{R}^n$, and this completes the proof. $\square$

**Lemma 5.** *If $F$ is a real analytic vector valued function that yields a bounded Koopman operator, $\mathcal{K}_F$, over the Gaussian RBF's native space, then its extension to an entire function, $F : \mathbb{C}^n \to \mathbb{C}^n$ induces a bounded operator over $H_G(\mathbb{C}^n)$.*

*Proof.* Since an entire function on $\mathbb{C}^n$ is uniquely determined by its restriction to $\mathbb{R}^n$, it follows that the span of the complex valued Gaussian RBFs with centers in $\mathbb{R}^n$ is dense in $H_G$. Moreover, to demonstrate that $\mathcal{K}_F$ is bounded, it suffices to show that there is a constant $C$ such that

$$C^2 K_G(z, w) - K_G(F(z), F(w)) \tag{11}$$

is a positive kernel. By the above remark, it suffices to show this for real $x, y \in \mathbb{R}^n$, but then this is equivalent to the statement that $\mathcal{K}_F$ is bounded over the Gaussian RBF's native space over $\mathbb{R}^n$. $\square$

**Theorem 2.** *If $F : \mathbb{C}^n \to \mathbb{C}^n$ is an entire function, and $\mathcal{K}_F$ is bounded on $H_G$, then $F(z) = Az + b$ for a matrix $A \in \mathbb{C}^{n \times n}$ and vector $b \in \mathbb{C}^n$.*

*Proof.* If $\mathcal{K}_F$ is bounded, then it has a bounded adjoint, $\mathcal{K}_F^*$, which acts on the complex Gaussian as $\mathcal{K}_F^* K_G(\cdot, z) = K_G(\cdot, F(z))$. In particular, there is a constant $C > 0$ such that $\|K_G(\cdot, F(z))\|_{H_G}^2 \leq C^2 \|K_G(\cdot, z)\|_{H_G}^2$. Noting the identity $\|K_G(\cdot, z)\|_{H_G}^2 = \exp\left(2 \sum_{j=1}^n (\Im z_j)^2\right)$ and taking the logarithm, it follows that

$$\sum_{j=1}^n (\Im F_j(z))^2 \leq \log(C^2) + \sum_{j=1}^n (\Im z_j)^2 \leq \log(C^2) + \|z\|_2^2. \tag{12}$$

From this inequality, it follows that for each coordinate $j = 1, \ldots, n$, the harmonic function $v_j(z) = \Im F_j(z)$ has linear growth. That is, there is a constant $\tilde{C}$ so that $|v_j(z)| \leq \tilde{C}(1 + \|z\|_2)$ for all $z \in \mathbb{C}^n$. It follows (e.g. from the standard Cauchy estimates) that $v_j(z) = v_j(x + iy)$ must be an affine linear function of $x$ and $y$, and therefore, so must its harmonic conjugate $u_j(z)$, and we conclude that $F(z) = Az + b$. $\square$

## C    Further Numerical Examples

### C.1    The Duffing oscillator

In this numerical example, sixteen $10-$seconds long trajectories of the Duffing oscillator, given by $\dot{x}_1 = x_2$ and $\dot{x}_2 = -0.1x_2 + x_1 - x_1^3$, are generated starting from initial states on a uniform grid on the set $[-1, 1] \times [-1, 1]$. The trajectories are sampled every 0.5 seconds to generate a set of snapshots. The snapshots are used to build a predictive model of the oscillator using the method developed in this paper (implemented using regularized regression with $\epsilon = 10^{-6}$ and the exponential dot product kernel with $\mu = 20$) and using the baseline method from (Williams et al., 2015b) (implemented using a threshold of $10^{-3}$ for zeroing singular values and the exponential dot product kernel with $\mu = 20$). To evaluate the predictive models, $15-$seconds long trajectories of the oscillator starting from 100 initial conditions randomly selected from the set $[-1.5, 1.5] \times [-1.5, 1.5]$ are generated by integrating the continuous-time model in equation 9 using a variable time-step fourth order Runge-Kutta integrator. These trajectories are compared against the true trajectories starting from the same initial conditions, generated by integrating the true dynamics using the same integrator. The metric used for comparison is the pointwise relative error $e(t) = \frac{\|x(t) - \hat{x}(t)\|}{\max_t \{\|x(t)\|\}}$, where $x(\cdot)$ is the true trajectory and $\hat{x}(\cdot)$ is the predicted trajectory. Figure 3 shows the pointwise mean of the relative error across the 100 trajectories along with the spread of the relative error from the pointwise minimum to the pointwise maximum.

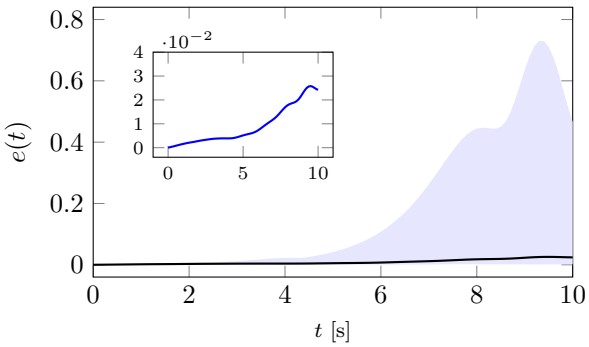

(a) Method developed in this paper, reconstruction using the continuous-time model in equation 9.

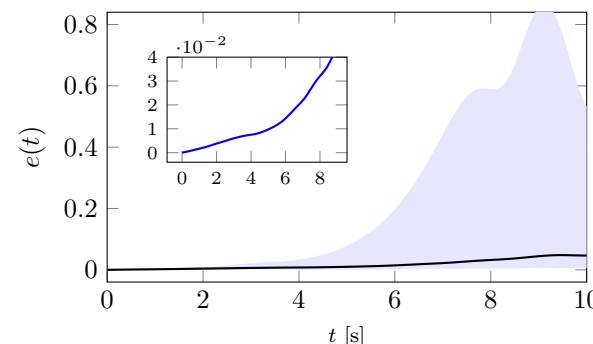

(b) Kernel DMD from (Williams et al., 2015b), reconstruction using the continuous-time model in equation 9.

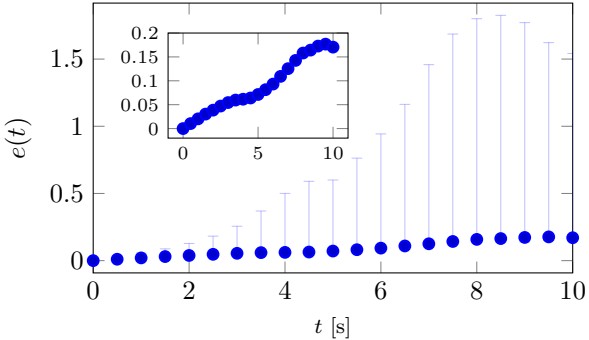

(c) Method developed in this paper, reconstruction using the discrete-time model in equation 8.

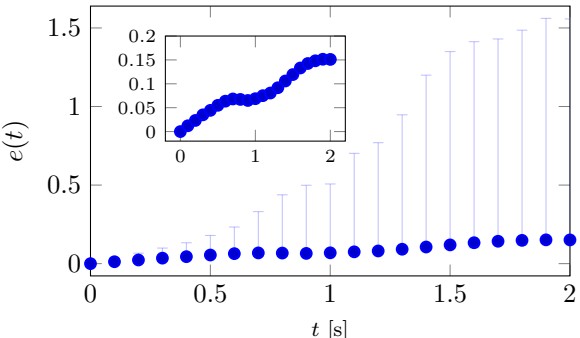

(d) Kernel DMD method in (Williams et al., 2015b), reconstruction using the discrete-time model in equation 8.

Figure 3: For the Duffing oscillator example, this figure shows the pointwise mean of the relative error between the true trajectories and the trajectories generated using DMD. The shaded area shows the pointwise maximum and the pointwise minimum over 100 randomly initialized trajectories.

To demonstrate the effectiveness of the heuristic continuous-time model in equation 9, the true trajectories, sampled every 0.5 seconds, are also compared against trajectories generated using the discrete-time model in equation 8. Figure 3 also shows the pointwise mean of the relative error across the 100 trajectories along with error bars that show the pointwise maximum and the pointwise minimum relative error.

## C.2 Turbulent flow

Another Numerical example uses a data set that accompanies (Jakob et al., 2020), which consist of flow simulations that range from laminar flow configurations to turbulent flow configurations. The data can be accessed from their website (Jakob et al., 2021). The flow data used in this paper (id number 7999) corresponds to 2-dimensional turbulent flow with a Reynolds number of 4092.216 and a Kinematic viscosity of $5.45 \exp -05$. The data consist of 1001 snapshots, each snapshot contains flow velocities in the horizontal and vertical directions measured on a regular grid of size $512 \times 512$ in two dimensions with domain $[0, 1] \times [0, 1]$. The snapshots are separated by 0.01 seconds. In this demonstration, snapshots 1 through 300 are used as the input data, and snapshots 2 through 301 are used as output data, assuming that the $i$th and $(i+1)$th snapshots satisfy $x_{i+1} = F(x_i)$ for some unknown nonlinear function $F$. The snapshots are normalized so that the largest 2-norm among all snapshots is 1.

The Koopman modes, eigenvalues, and eigenfunctions are then computed using the developed technique and snapshots 2 through 301 are reconstructed from snapshot 1 using equation 7. DMD is implemented using the Gaussian RBF kernel, $K(x, y) = \exp\left(-\frac{1}{\mu}\|x - y\|_2^2\right)$ (with $\mu = 0.75$ and $\epsilon = 0.0001$). The baseline method

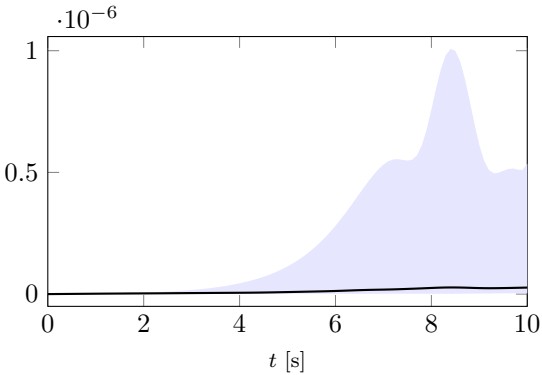

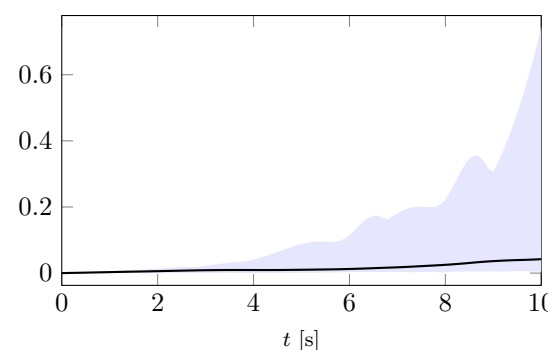

(a) Difference between prediction generated using the method in (Williams et al., 2015b) and the truncated pseudoinverse implementation of the method developed in this paper.

(b) Difference between trajectories generated using the method in (Williams et al., 2015b) and the regularized regression implementation of the method developed in this paper.

Figure 4: For the Duffing oscillator example, this figure shows the pointwise mean of the error between the trajectories predicted using the continuous-time model in equation 9 generated by the method in (Williams et al., 2015b) and the method developed in this paper. The shaded area shows the pointwise maximum and the pointwise minimum over 100 randomly initialized trajectories.

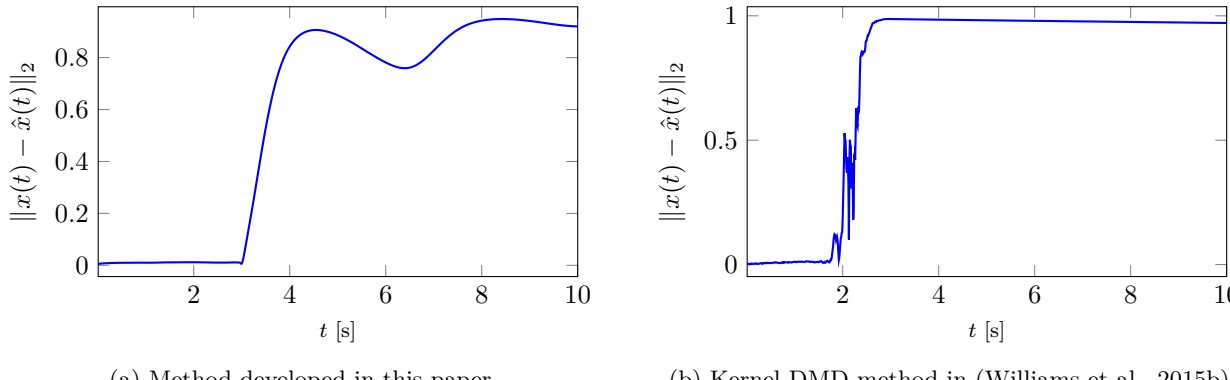

(a) Method developed in this paper

(b) Kernel DMD method in (Williams et al., 2015b)

Figure 5: For the turbulent flow example, this figure shows the 2-norm of the error between the true snapshots and the snapshots generated using DMD.

from (Williams et al., 2015b) is implemented using the same kernel with $\mu = 0.000518$. The predictive model in equation 7 is also used to generate 700 additional snapshots.

## C.3 Additional Discussion

The ability of the developed DMD technique to reconstruct the training data from the same initial condition is apparent from figures 3 and 5. In the duffing oscillator example the pointwise mean of the relative error is shown to be nearly zero. The relative reconstruction errors for the turbulent flow are of the order of $1e - 2$.

The duffing oscillator example, is also used to highlight the similarity in predictions generated by the DMD method developed in this manuscript and that proposed by (Williams et al., 2015b). Figure 4 shows that the difference between the predictions generated by a truncated pseudoinverse implementation of the method developed in this paper and the predictions generated by the method in (Williams et al., 2015b) are of the order of $1e - 6$. As expected, figure 4 further highlights that choosing to use the truncated pseudoinverse implementation as opposed to the regularized regression implementation results in predictions that are more similar to those obtained under the method in (Williams et al., 2015b). Furthermore, figure 3 also shows

that for the duffing oscillator example the continuous-time model in equation 9 results in a smaller pointwise mean relative error than the discrete-time model in equation 8.

In the case of the turbulent flow, as shown in 5, the relative prediction errors for the method developed in this paper and the baseline kernel DMD method developed in (Williams et al., 2015b) quickly increase to 1. Since the data are normalized so that the largest 2-norm among all snapshots is 1, a relative prediction error of 1 signifies a very poor match between the predicted state and the true state. Such performance degradation is expected for turbulent flows since the underlying assumption that the $i$th and $(i + 1)$th snapshots satisfy $x_{i+1} = F(x_i)$ for some unknown nonlinear function $F$ is unlikely to generalize outside the training data when the flow is turbulent. From figure 5, the method developed in this paper appears to perform slightly better than the baseline. This performance improvement can be attributed to the explicit regularization of the Gram matrix, where the regularization parameter can be tuned in addition to the kernel width parameter. in (Williams et al., 2015b), the regularization is done implicitly through the pseudoinverse operation. We postulate that by manipulating the threshold used to decide which singular values are zero in the pseudoinverse computation, the performance of the kernel DMD method can be improved.

