# OpenReview forum: "The Kernel Perspective on Dynamic Mode Decomposition"
_TMLR — Accepted by TMLR_

### Review · Reviewer_fD6Z · 2024-04-24

**Summary Of Contributions:**

An algorithm similar (or almost identical) to the kernel DMD is derived in a perspective different from previous studies. Specifically, the authors start at the projection of the Koopman operator onto a set of basis functions comprising kernel functions centered at the snapshots. Although the resulting algorithm seems to be basically identical to what is known as kernel DMD (eg WIlliams+ 2015), the contribution of this work is claimed to be its derivation from the given perspective. Moreover, some analyses of the Koopman operator in RKHS and its eigenfunctions are presented.

**Audience:**

Yes

**Claims And Evidence:**

Yes

**Requested Changes:**

(1) As mentioned above, please elaborate, if any, on the connection between the result in Section 4.2.3 and that in Ikeda et al. (2022).

(2) The difference between the two algorithms used in Section 7.2 could be clearer.

**Strengths And Weaknesses:**

### Strengths

+ The derivation of "kernel DMD" from the perspective of finite rank representation of the Koopman operator sounds indeed interesting.

+ The writing is not bad, I could follow the main reasoning though I am not well equipped with such mathematical discussions.

### Weaknesses

- That being said, I feel that the relationship between the claims scattered in Sections 4-6 could have been clearer. A short paragraph on how they relate to each other might help.

- The numerical examples do not seem wrong, but it is unclear how they are valuable. I am not following the discussions that happened in the previous submission, but my first impression is that they were added just to make the paper slightly more ML-ish. A possibility I would raise (which I am not recommending but just showing) is to defer the numerical examples to an appendix to fit the paper within the nominal 12-page length.

- How the discussions in Sections 4.1, 4.2.1, and 4.2.2 relate to the discussion afterward in Sections 5 and 6 is unclear.

- The novelty of the result in Section 4.2.3 is unclear. For example, could you please clarify the connection to Ikeda et al. (2022), if any?

[Ikeda et al., 2022] Ikeda et al., Boundedness of composition operators on reproducing kernel Hilbert spaces with analytic positive definite functions, Journal of Mathematical Analysis and Applications 511:126048, 2022.

---

> ### Author Response · Authors · 2024-05-14
> **Response to Reviewer fD6Z**
>
> As reviewer fD6Z and the others have pointed out, a major weakness in our
> presentation is clarity. We will endeavor to correct this in our revision, specifically
> addressing the claims in Sections 4 - 6.
>
> The numerical examples are really just demonstrating a proof of concept here,
> since the contributions of the manuscript are mostly theoretical. However, since
> the other reviewers have requested that we add more experiments, we will move
> the experiments to an appendix as fD6Z suggests.
>
> We were not aware of Ikeda’s work when we drafted this manuscript. Indeed, we
> will include Ikeda in the revisions of this work. We do contribute a different proof
> of the same result, however, where Ikeda leverages the Fourier definition of the
> Gaussian RBF kernel, and we instead leverage its connection with the Fock space. So our proof is novel, even if it appears the ultimate result has already been
> established. We will make the appropriate adjustments to make sure that Ikeda
> receives the proper credit.

---

> > ### Comment · Reviewer_fD6Z · 2024-05-31
> >
> > Thanks for the response. It sounds convincing but I cannot find any updates in the manuscript.
> >
> > In my understanding, one of the points of TMLR's reviewing process is to update the manuscript according to reviews and discussions, just like in usual journal submissions, and unlike conferences. Although it is likely that the authors will indeed make the suggested changes, I think it also makes sense for me to wait to send a recommendation until the authors submit a revision.

---

### Review · Reviewer_64sb · 2024-04-26

**Summary Of Contributions:**

The work is focused on characterizing the properties of Koopman operators (KO) over reproducing kernel Hilbert spaces (RKHS), as a way to bring forth a somewhat novel Kernel perspective to $Koopmanism$ - contrasting it with the usual ergodic picture. A central result for example showcases how the class of bounded Koopman operators over RKHS is miniscule even when the considered dynamics is being governed by analytic maps - an additional assumption that the map be affine is critical to impart enough structure to support bounded KOs (and even then in a restricted Guassian RBF setting).

However, while seemingly a very strong restriction on the viability of $Koopmanism$ (the lack of compactness and/or boundedness in operators often makes functional analytic problems completely intractable), within an assumption of densely defined KOs (amongst other smaller ones), we find that enough structure is still preserved to make it numerically viable, making this an impressive attempt at providing a novel, kernel based theoretical perspective on DMD.

The proposed theory leads to some natural algorithms, which are tested for their advantages and then compared with existing methods to demonstrate equivalence.

**Audience:**

Yes

**Claims And Evidence:**

Yes

**Requested Changes:**

On a technical/mathematical level, I have no issues and have found no substantial errors.

On a reading front, I think the authors have struggled with the intersectional nature of this work - writing for an ML audience while needing to leverage fairly standard but still bulky and involved mathematical machinery. My personal advice would be to write every novel result/deduction/argument out in a standard mathematical format.

Where arguments are being developed, put them into the service of proving or conjecturing a precise statement. Where claims are being made, always state them as precise statements/lemmas/theorems. Everytime a definition is stated, it should be as an indexed object that can be referred to instantly by simply looking at the right definition number, rather than as a sentence in some hard to find paragraph. Etc. Etc.

I don't mean to suggest that the above approach is completely absent: I am strongly recommending that this approach be applied to every mathematical portion of this paper. To further simplify reading, the authors can then pick and choose the most important statements and lemmas to build a storyline of precise statements and implications in the "main text" and then put everything else (including all the proofs) in a well-referenced appendix. This way, people who care can easily access full technical details while the rest can follow the important portions of the story without needing to untangle the entire technical machinery.

In summary, the work is accurate and convincing but highly unclear: I can't recommend acceptance in good faith until that last point is resolved. However, I would again strongly acknowledge for the editors that I have no technical changes requested or errors to point out here - if the struggles in reading were simply mine, please feel free to completely discount the above.

**Strengths And Weaknesses:**

Strengths:
1. To my knowledge, a somewhat novel, kernel based perspective on DMD/Koopmanism has been promulgated in this work, along with a more impressive set of counterexamples to suggest why Koopmanism isn't necessarily well suited for anything beyond affine problems, before further analysis provides a strong counter-counter-argument as to why the situation is not critically deficient after all, if the KOs are considered to be densely defined. If the reader can slog through the text (see weakness 1), the proposed theory is an interesting and fruitful way to consider the existing empirical literature in these directions and a stark departure from the ergodic perspective that heavily dominates how Koopmanism is formalized usually.

2. To further bolster the value of their case, the authors investigate some naturally implied algorithms within the perspective and somewhat validate that their results are equivalent to existing methods (see weakness 2).

Weaknesses:
1. I don't mean to be harsh, but the work is badly-written - I would consider myself well versed with most of the involved subject matter and it still took me an unreasonable amount of time to simply figure out when known facts were being repeated and when something novel was being added. It is even more confusing why this should be the case since in my opinion, the novel contributions are truly exciting and remarkable. A lot of the mathematical arguments, results, and proofs have been presented in an almost stream of consciousness style that is not amenable to reading about such dense technical matters.

2. Numerical analysis is not insufficient, but also far from conclusive. It is also presented in an sub-optimal manner. For ex, in both Fig. 7.1 and 7.3, the relevant quantity of interest is not the outputs of the two methods, but the difference between the two method outputs (I am not suggesting the removal of those figures, but explicitly the quantity of interest is the difference between those two sub-figures in each case - that comparison should be present). I would also be more comfortable if at least a couple of more experimental verifications were included, rather than just 2.

---

> ### Author Response · Authors · 2024-05-14
> **Response to Reviewer 64sb**
>
> Thank you for your feedback, and we are happy that you find our results exciting.
> We will endeavor to give a cleaner presentation in the revised manuscript, where we will be much more explicit about pointing out our novel contributions.
>
> We will also include several additional experiments to demonstrate the ability of
> our method. However, we do emphasize that the contributions of this manuscript
> are mostly theoretical.

---

> ### Comment · Reviewer_64sb · 2024-08-26
> **Comments on the new version**
>
> I have reviewed the updated version of the document, which now meets the standard for acceptance on the presentation/clarity front (I will note that even now, there is scope for significant improvement on this front). I can recommend the work for acceptance with these updates.

---

### Review · Reviewer_ot35 · 2024-05-01

**Summary Of Contributions:**

The paper proposes a theoretical contribution related to the study of Koopman operators over RKHSs. It proves that if the Koopman operator of a real entire vector-valued function $F$ is bounded over the Gaussian RBF space, then $F$ is affine.

Furthermore, the paper discusses the feasibility of the assumptions that are often made about the Koopman operator and that may not be relevant when they are considered on RKHSs, arriving at the conclusion that they cannnot be assumed to be bounded and compact.

The paper then proposes to work in the context of densely defined operators and establishes a DMD method in this case that closely mimics previous work about kernel DMD. Numerical experiments highlight the benefit of the approach.

**Audience:**

Yes

**Claims And Evidence:**

Yes

**Requested Changes:**

- Improve the presentation of the differences with previous work [critical]
- Better discuss the assumptions (densely defined + identity in the RKHs) [critical]

**Strengths And Weaknesses:**

Strenghts

- The paper is of interest for the TMLR community. The theoretical contribution is sound and readers might be interested in the characterization of bounded Koopman operators over Gaussian RKHSs.

- The numerical experiments complement nicely the theoretical derivations.

Weaknesses:
- The paper questions many assumptions regarding Koopman operators but leaves some out of questionning. While I understand the interest of looking at a Koopman operator as an operator from $H \to H$, it introduces a main question that needs addressing in my opinion: "how big is $\{g \in H | g \circ F \in H \}$" ? Is the densely defined assumption really reasonnable for say the Gaussian RBF kernel ? What type of $F$ are needed to have $\forall x \in X, y \mapsto k(F(y), x) \in H $ ?

- The assumption "$x \mapsto x$ is in $H$" is a heavy one and must be better highlighted. What happens with the gaussian kernel which violates this assumption ?

- The relation to [Williams 2015a] is not well explicited in my opinion. From what I understand, both methods yield the same matrix whose eigendecomposition is used to approximate the dynamics. Is the difference only regarding the assumptions made on the Koopman operator ? Or is it in the regularization performed when the matrix is not invertible ? Or further down the line when computing the modes ?

Additional remarks:
- In the text before equation (5), when writing the decomposition of $(x)_d$, there is a typo on the index: $\phi_i$ should be $\phi_j$.

---

> ### Author Response · Authors · 2024-05-14
> **Response to Reviewer ot35**
>
> The collection of densely defined operators depends critically on the selected
> space. To determine how broad this is in the case of the Gaussian RBF in
> particular is difficult, and perhaps worth exploring as another paper. However,
> over many spaces of analytic functions, we can answer that question very quickly.
>
> For instance, over the Fock space, we know that the collection of polynomials is
> dense, and compositions of polynomials with other polynomials produces
> polynomials. That means if our composition symbol is a polynomial, then the
> corresponding Koopman operator is densely defined.
>
> Whether or not the identity function is in the space is indeed important. For the
> Fock space, this certainly holds, but it does not hold for the Gaussian RBF. On the
> other hand, the Gaussian RBF’s native space is universal. Hence, over any given
> compact workspace, we can find a function inside of the space that estimates the identity function accurately and uniformly over that workspace. One ad hoc
> approach would be to artificially add the full state observable to the Hilbert space
> itself by making a new kernel x^Ty + K(x,y).
>
> It’s not the ideal situation, and indeed, the theory is better founded over spaces
> like the Fock space, which have the full state observable within the space.
>
> In truth, the assumptions made by the Williams et al paper on the Koopman
> operator are implicit. They work almost exclusively with feature spaces, which
> obscures the overall theory. The two matrices produced, after considerable
> manipulation of infinite dimensional matrices, can be seen in equation (16) of their
> paper. These matrices are where the eigendecompositions actually happen. We
> arrive at the same two matrices through Hilbert space reasoning, where we can
> be more explicit with the assumptions on the operator, and without appealing to
> infinite dimensional matrices

---

> > ### Comment · Reviewer_ot35 · 2024-05-30
> > **Acknowledging rebuttal**
> >
> > Thank you for your answer. I have no further question.

---

### Public Comment · ~Max_Beier1 · 2024-06-18
**A Comment on the State of the Art of Dynamic Mode Decomposition in RKHS**

## Summary
The state of the art of Koopman (Transfer) Operator Regression is no longer Williams et al. with kDMD or eDMD. Work has been done to explain DMD from a kernel view; it should be addressed to ensure high machine-learning standards.

Currently, the paper does not reflect the field it contributes to and, as such, misses the main open challenges in DMD/ transfer operator regression. This might be misleading for newcomers to the field.

## RKHS-based operator learning
[Learning Dynamical Systems via Koopman Operator Regression in Reproducing Kernel Hilbert Spaces, Neurips 2022][1]:
- The estimator from Algorithm 1 is derived as the solution to a ridge regression problem in RKHS (p. 6). The referenced approach has the benefit of an *interpretable* RKHS norm regularizer.
- For well-posedness, self-adjointness is assumed, and they derive the bounded HS-norm of the transfer operator.

## Operator-theoretic learning in ergodic and non-stationary settings
[Learning invariant representations of time-homogeneous stochastic dynamical systems, ICLR 2024][2]:
- Applies general PDE and operator theory knowledge. I.e., assuming compactness of transfer operators of stochastic systems is a non-issue, this requires an invariant distribution, which often exists.
- Operator-theoretic properties specific to dynamical systems are explored. They prove to be important in practice and algorithmically controllable.

## Operator-based learning in non-ergodic settings
[Koopman Kernel Regression, Neurips 2023][3]:
- There has been work on ensuring boundedness for deterministic systems. The main tool I am aware of is through control over the domain of the flow $F$ and thus of the basis functions $\psi$. A compact, non-recurrent domain solves boundedness issues and yields well-posed estimators.
- The apparent downside of this assumption is that the analysis of invariants is restricted to that domain. Finite-time invariants are uncountable.
- On such a non-recurrent domain, operator regression problems can be posed in a well-defined manner.

### Affine Dynamics:
The proof that this is the case is in argument similar to the recently published (but long common knowledge) [Reproducing kernel Hilbert spaces cannot contain all continuous functions on a compact metric space][4]. The gist of this note is: RKHS are regular. The space of continuous functions C(X) has (much) less structure. Yet RKHS are incredibly useful to approximate continuous functions in a sampling-based fashion (via. Stone-Weierstrass). The remedy for exact representations are RKBS, which are currently under active development. A similar constructive outlook is missing in the paper, but it would, IMO be *the most interesting* takeaway for the community.



[1]:https://proceedings.neurips.cc/paper_files/paper/2022/file/196c4e02b7464c554f0f5646af5d502e-Paper-Conference.pdf

[2]:https://arxiv.org/pdf/2307.09912

[3]:https://proceedings.neurips.cc/paper_files/paper/2023/hash/34678d08b36076de986df95c5bbba92f-Abstract-Conference.html

[4]:https://link.springer.com/article/10.1007/s00013-024-01976-0

---

> ### Author Response · Authors · 2024-06-19
> **Re: Max Beier [Part 1]**
>
> Hello Dr. Beier,
>
> We very much appreciate the manuscripts you have pointed us to here. We will do our best to include them in the final version of our manuscript. In the meantime, we would like to leverage this list to further emphasize the importance of our work here.
>
> The manuscript, “Learning Dynamical Systems via Koopman Operator Regression in Reproducing Kernel Hilbert Spaces,” (Neurips 2022) is exceptionally well written and I thoroughly enjoyed looking at their perspective on the problem. There are several things I would like to point out concerning the assumptions made in their manuscript.
>
> As you have pointed out, they have made the assumption that the Koopman operator is self adjoint. The authors also mention that while they do make this assumption, that even normality of the Koopman operator cannot be guaranteed. Indeed, in “Liouville operators over the Hardy space” (Journal of Mathematical Analysis and Applications 2022) it was shown that the assumption of self adjointness can dramatically restrict the dynamics allowed in that situation. Indeed, the latter manuscript showed that only dynamics that admit a self adjoint densely defined Koopman generator over the Hardy space (which fits in the framework of the NeurIPS paper) are those dynamics of the form $\dot x = c x$ for real $c$.
>
> The assumptions concerning compactness of the Koopman operator have already been addressed in our manuscript. Certainly, some Koopman operators can be compact, but that depends on the selection of the Hilbert space, and we demonstrate here that for many commonly used spaces, the only dynamics for which this actually works are affine.
>
> Finally, the NeurIPS 2022 paper actually cites an arXiv preprint of our manuscript. Which further emphasizes the value of this present work. The title of this paper differs from the version of our manuscript that the NeurIPS 2022 paper cites, but unfortunately, for the sake of anonymity, we can’t comment on this further.
>
> The second paper, titled “Learning Invariant Representations of Time-Homogenous Stochastic Dynamical Systems,” is an interesting approach to constructing a RKHS from data, and reminds me somewhat of Kawahara’s NeurIPS paper which learned a RKHS via Krylov methods in that both seek to determine an empirical representation of a RKHS over which the Koopman operator may be bounded or compact.
>
> We will have to give more thought to this approach and see if we can connect it with our discussion in the our manuscript. It is a very interesting approach which uses Deep Neural Networks to build a new Hilbert space.
>
> We would like to point out that this second manuscript was published in a conference in 2024, while the first version of our manuscript we submitted to TMLR was in 2023, and the arXiv version of our manuscript has been available since 2021. I hope you can appreciate that we could hardly know about new theoretical developments that appear after we submitted our manuscript.
>
> Moreover, the arXiv version of our manuscript was cited in the first paper in your list, which in turn was cited by this second manuscript. Again, emphasizing the merit of our manuscript to see publication.
>
>
> The third manuscript appears to be your own.
> I think it is an interesting result and is very similar to ”The Occupation Kernel Method for Nonlinear System Identification” published recently in SICON (but available on arXiv since 2019). There the derived space uses gradients on the kernel function, but also have the double integrals over the trajectories, and was made to correspond well with the Koopman generator. It appears that this third paper overlooked that result.
>
> This Hilbert space construction in the third paper is again empirically derived, and the invariants are not actually guaranteed to be in this space. Without a full description of this Hilbert space, it is difficult to declare what Koopman operators are actually permitted to be bounded and compact. In fact, boundedness is an assumption made in the paper, rather than something that is proved.
>
> [Continued in Part 2]

---

> ### Author Response · Authors · 2024-06-19
> **Re: Max Beier [Part 2]**
>
> [Continued from Part 1 - Character limits got in the way]
>
> To your last point, concerning universality. I would like to emphasize that the Hilbert space is different from the collection of symbols (or dynamics) of the Koopman operators over that Hilbert space. Just because you have a universal RKHS, it does not follow that the collection of symbols that admit a bounded Koopman operator are going to be universal themselves. In fact, this manuscript presents several examples where the dynamics are affine. In fact, in this very discussion, we showed that there is at least one case where the assumption of symmetry leaves only a one dimensional space of symbols, $cx$.
>
> The consideration of symbols of operators over RKHSs is a very well studied area. As mentioned in the paper, Carswell et al showed that the only Koopman operators (composition operators) over the Fock space that are bounded are those with affine symbol. This was also shown for the Paley Weiner space and easy enough to work out by hand for the polynomial space.
>
> Broadening out from composition operators, even multiplication operators can demonstrate similar restrictiveness, where the only bounded multiplication operators over the Fock space are those with constant symbols. And changing the conditions on the operators either broadens or shrinks the collection of admissible symbols. The field of Function Theoretic Operator Theory is dedicated to connecting various operator theoretic notions, boundedness, compactness, self-adjointness, etc with their impact on the functions or symbols that represent those operators.
>
> In this case, boundedness and compactness yield very few admissible symbols for Koopman operators, and this manuscript provides a numerical routine that is theoretically well founded even in the absence of the boundless condition on the Koopman operators.
>
> Thank you again for taking the time to read our manuscript and to provide your comments. This exemplifies the value of the open review system, where we can have an engaging discussion and clarify the points in our manuscript.  The papers you provided are genuinely enlightening and we look forward to seeing more of your own work.
>
> Sincerely,
>
> Anonymous Authors

---

### Decision · Action_Editor_Jczv · 2024-08-26

**Recommendation:** Accept as is

**Comment:**

All reviewers recommend acceptance and I believe the merits of the paper (the novel theoretical perspective) is worth being published in TMLR.

The authors spent considerable amount of time revising their manuscript and the improvement in clarity was noted by reviewers. None of the reviewers provided additional (specific) points that the authors could address in another round of revision, so I am recommending acceptance as is.

**Audience:**

Yes. There is a growing presence of Koopman operator theory in the machine learning community, and this work provides new theoretical perspectives that would be of interest to those working on developing new theory and methods for Koopman operator theory.

**Claims And Evidence:**

All reviewers agreed that the theorems in the paper were technically sound and the numerical results (while weaker than current state of the art) were done well to support the theoretical perspective of the paper.

All reviewers commented that the authors could continue to improve the clarity of their results, but that they were acceptable as is.